# Kernel Interpolation with Sparse Grids

**Mohit Yadav**
University of Massachusetts Amherst
ymohit@cs.umass.edu

**Daniel Sheldon**
University of Massachusetts Amherst
sheldon@cs.umass.edu

**Cameron Musco**
University of Massachusetts Amherst
cmusco@cs.umass.edu

## Abstract

Structured kernel interpolation (SKI) accelerates Gaussian process (GP) inference by interpolating the kernel covariance function using a dense grid of inducing points, whose corresponding kernel matrix is highly structured and thus amenable to fast linear algebra. Unfortunately, SKI scales poorly in the dimension of the input points, since the dense grid size grows exponentially with the dimension. To mitigate this issue, we propose the use of *sparse grids* within the SKI framework. These grids enable accurate interpolation, but with a number of points growing more slowly with dimension. We contribute a novel nearly linear time matrix-vector multiplication algorithm for the sparse grid kernel matrix. We also describe how sparse grids can be combined with an efficient interpolation scheme based on simplices. With these modifications, we demonstrate that SKI can be scaled to higher dimensions while maintaining accuracy.

## 1 Introduction

Gaussian processes (GPs) are popular prior distributions over continuous functions for use in Bayesian inference [14]. Due to their simple mathematical structure, closed form expressions can be given for posterior inference [20]. Unfortunately, a well-established limitation of GPs is that they are difficult to scale to large datasets. In particular, for both exact posterior inference and the exact log-likelihood computation for hyperparameter learning, one must invert a dense kernel covariance matrix $K \in \mathbb{R}^{n \times n}$, where $n$ is the number of training points. Naively, this operation requires $\mathcal{O}(n^3)$ time and $\mathcal{O}(n^2)$ memory.

**Structured Kernel Interpolation.** Many techniques have been proposed to mitigate this scalability issue [23, 19, 24, 9]. Recently, structured kernel interpolation (SKI) has emerged as a promising approach [9]. In SKI, the kernel matrix is approximated via interpolation onto a dense rectilinear grid of $m$ inducing points. In particular, $K$ is approximated as $W K_G W^T$, where $K_G \in \mathbb{R}^{m \times m}$ is the kernel matrix on the inducing points and $W \in \mathbb{R}^{n \times m}$ is an interpolation weight matrix mapping training points to nearby grid points. Typically, $W$ is sparse, and $K_G$ is highly structured — e.g., for shift invariant kernels, $K_G$ is multi-level Toeplitz. Thus, $W$, $K_G$, and in turn the approximate kernel matrix $W K_G W^T$ admit fast matrix-vector multiplication. This allows fast approximate inference and log-likelihood computation via the use of iterative methods, e.g., the conjugate gradient algorithm.

**SKI's Curse of Dimensionality.** Unfortunately, SKI does not scale well to high-dimensional input data: the number of points in the dense grid, and hence the size of $K_G$, grows exponentially in the dimension $d$. Moreover, SKI typically employs local cubic interpolation, which leads to an interpolation weight matrix $W$ with row sparsity that also scales exponentially in $d$. This *curse of*

*dimensionality* is a well-known issue with the use of dense grid interpolation. It has been studied extensively, e.g., in the context of high-dimensional interpolation and numerical integration [17, 5].

In the computational mathematics community, an important technique for interpolating functions in high dimensions is *sparse grids* [2]. Roughly a sparse grid is a union of rectilinear grids with different resolutions in each dimension. In particular, it is a union of all $2^{\ell_1} \times 2^{\ell_2} \times \ldots \times 2^{\ell_d}$ sized grids, where $\sum_{i=1}^{d} \ell_i \leq \ell$, for some maximum total resolution $\ell$. This upper bound on the total resolution limits the number of points in each grid — while a grid can be dense in a few dimensions, no grid can be dense in all dimensions. See Figure 1 for an illustration. Sparse grids have interpolation accuracy comparable to dense grids under certain smoothness assumptions on the interpolated function [22], while using significantly fewer points. Concretely, for any function $f : \mathbb{R}^d \to \mathbb{R}$ with bounded mixed partial derivatives, a sparse grid containing $\mathcal{O}(2^\ell \ell^{d-1})$ points can interpolate as accurately as a dense grid with $\mathcal{O}(2^{\ell d})$ points, where $\ell$ is the maximum grid resolution [21].

**Combining Sparse Grids with SKI.** Our main contribution is to demonstrate that sparse grids can be used within the SKI framework to significantly improve scaling with dimension. Doing so requires several algorithmic developments. When the inducing point grid is sparse, the kernel matrix on the grid, $K_G$, no longer has simple structure. E.g., it is not Toeplitz when the kernel is shift invariant. Thus, naive matrix-vector multiplications with $K_G$ require time that scales quadratically, rather than near-linearly in the grid size. This would significantly limit the scope of performance improvement from using a sparse grid. To handle this issue, we develop a near-linear[1] time matrix-vector multiplication (MVM) algorithm for any sparse grid kernel matrix. Our algorithm is recursive, and critically leverages the fact that sparse grids can be constructed from smaller dense grids and that they have tensor product structure across dimensions. For an illustration of our algorithm's complexity versus that of naive quadratic time MVMs, see Figure 1 (c).

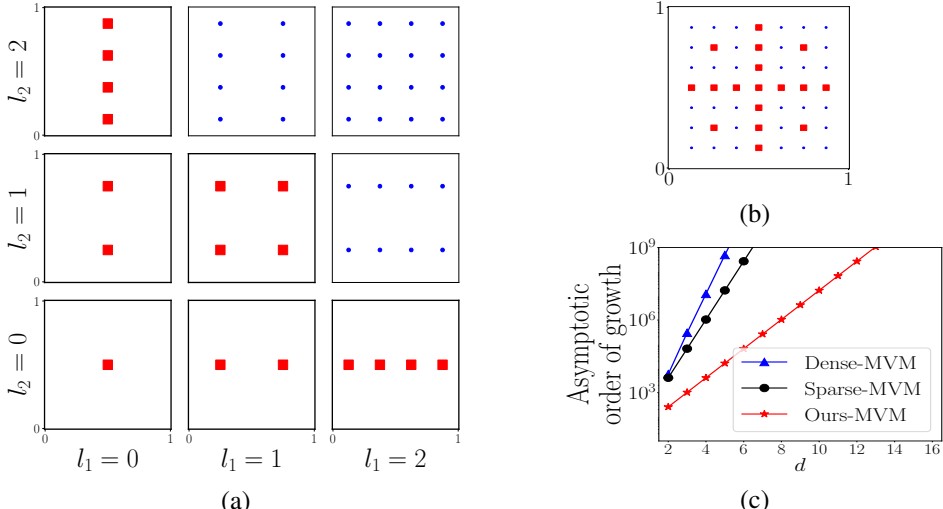

(a)  (b)  (c)

**Figure 1:** Illustration of sparse grid construction for $d = 2$ and maximum resolution $\ell = 2$. (a) A dense grid with resolution $(l_1, l_2)$ has $2^{l_1}$ and $2^{l_2}$ points in each dimension; the grids with red square points have total resolution $l_1 + l_2 \leq 2$. (b) The sparse grid $\mathcal{G}_{2,2}$ is the union of rectilinear grids with total resolution at most 2. The 17 points with red squares belong to the sparse grid. The dense grid includes the additional points with blue dots, for a total of 49 points. (c) The asymptotic order of growth in performing a single kernel MVM operation for both grids with $2^8$ unique points in each dimension, ignoring constants. For sparse grids, *Ours-MVM* (i.e., the proposed MVM algorithm) improves significantly over the naive implementation (*Sparse-MVM*).

A second key challenge is that, while sparse grids allow for a grid size that grows as a much more mild exponential function of the dimension $d$, the bottleneck for applying SKI on large datasets can come in computing MVMs with the interpolation weight matrix $W \in \mathbb{R}^{n \times m}$. For classic high-dimensional interpolation schemes, like cubic interpolation, each row of $W$ has $\mathcal{O}(2^d)$ non-zero entries, i.e., the kernel covariance for each training point is approximated by a weighted sum of the covariance at $\mathcal{O}(2^d)$ grid points. When the number of training points $n$ is large, storing $W$ in

---

[1]'Near-linear' here means running in time $\mathcal{O}(m \log m)$ for a sparse grid with $m$ points.

memory, and multiplying by it, can become prohibitively expensive. To handle this issue, we take an approach similar to that of Kapoor et al. [13] and employ simplicial basis functions for interpolation, whose support grows linearly with $d$. Combined with our fast MVM algorithm for sparse grid kernel matrices, this interpolation scheme lets us scale SKI to higher dimensions.

In summary, we propose the use of *sparse grids* to improve the scalability of kernel interpolation for GP inference relative to the number of dimensions. To this end, we develop an efficient nearly linear time matrix-vector multiplication algorithm for the sparse grid kernel matrix. Furthermore, we also propose the use of simplicial interpolation to improve scalability of SKI for both dense and sparse grids. We show empirically that these ideas allow SKI to scale to at least 10 dimensions and perform competitively with state-of-the art GP regression methods. We provide an efficient GPU implementation of the proposed algorithm compatible with GPyTorch [9], which is available at https://github.com/ymohit/skisg and licensed under the MIT license.

## 2 Background

**Notation.** We let $\mathbb{N}$ and $\mathbb{N}_0$ denote the natural numbers and $\mathbb{N} \cup \{0\}$ respectively. Matrices are represented by capital letters, and vectors by bold letters. $I$ denotes the identity matrix, with dimensions apparent from context. For a matrix $M$, $\mathrm{mvm}(M)$ denotes the number of operations required to multiply $M$ by any admissible vector.

In GP regression, the training data are modeled as noisy measurements of a random function $f$ drawn from a GP prior, denoted $f \sim \mathcal{N}(0, k(\cdot, \cdot))$, where $k : \mathbb{R}^d \times \mathbb{R}^d \to \mathbb{R}$ is a covariance kernel. For an input $\mathbf{x}_i$, the observed value is modeled as $\mathbf{y}_i = f(\mathbf{x}_i) + \epsilon_i$, with $\epsilon_i \sim \mathcal{N}(0, \sigma^2)$. Observed training pairs $(\mathbf{x}_i, y_i)$ are collected as $X = [\mathbf{x_1}, \dots, \mathbf{x_n}] \in \mathbb{R}^{n \times d}$ and $\mathbf{y} = [y_1, \dots, y_n] \in \mathbb{R}^n$. The kernel matrix (on training data) is $K_X = [k(\mathbf{x}_i, \mathbf{x}_j)]_{i,j=1}^n \in \mathbb{R}^{n \times n}$. The GP inference tasks are to compute the posterior distribution of $f$ given $(X, \mathbf{y})$, which itself is a Gaussian process, and to compute the marginal log-likelihood $\log p(\mathbf{y})$. Naive approaches rely on the Cholesky decomposition of the matrix $\overline{K}_X = K_X + \sigma^2 I$, which takes $\Theta(n^3)$ time; see Rasmussen [20] for more details.

To avoid the $\Theta(n^3)$ running time of naive inference, many modern methods use iterative algorithms such as the conjugate gradient (CG) algorithm to perform GP inference in a way that accesses the kernel matrix only through matrix vector multiplication (MVM), i.e., the mapping $\mathbf{v} \mapsto \overline{K}_X \mathbf{v}$ [9]. These methods support highly accurate approximate solutions to GP posterior inference task as well as hyper-parameter optimization. The complexity of posterior inference and one step of hyper-parameter optimization is $\Theta(pn^2)$ for $p$ CG iterations, as $\mathrm{mvm}(\overline{K}_X) = n^2$. In practice, $p \ll n$ suffices [27, 9].

### 2.1 SKI: Structured Kernel Interpolation

SKI further accelerates iterative GP inference by approximating the kernel matrix in a way that makes matrix-vector multiplications faster [27]. Given a set of inducing points $U \subset \mathbb{R}^d$, SKI approximates the kernel function as $\tilde{k}(\mathbf{x}, \mathbf{x}') \triangleq \mathbf{w}_\mathbf{x}^T K_U \mathbf{w}_{\mathbf{x}'}$, where $K_U \in \mathbb{R}^{|U| \times |U|}$ is the kernel matrix for the set of inducing points $U$, and the vector $\mathbf{w}_\mathbf{x} \in \mathbb{R}^{|U|}$ contains interpolation weights to interpolate from $U$ to any $\mathbf{x}$. The SKI approximate kernel matrix is $\tilde{K}_X = W K_U W^T$, where $W \in \mathbb{R}^{n \times |U|}$ is the matrix with $i^{th}$ row equal to $\mathbf{w}_{\mathbf{x}_i}$.

To accelerate matrix-vector multiplications with the approximate kernel matrix, SKI places inducing points on a regular grid and uses grid-based interpolation. This leads to a sparse interpolation weight matrix $W$—for example, with cubic interpolation there are $O(4^d)$ entries per row, so that $\mathrm{mvm}(W) = O(n4^d)$—and to a kernel matrix $K_U$ that is multi-level Toeplitz (if $k$ is stationary) [27], so that $\mathrm{mvm}(K_U) = O(|U| \log |U|)$. Overall, $\mathrm{mvm}(\tilde{K}_X) = O(n4^d + |U| \log |U|)$, which is much faster than $n^2$ for small $d$. However, SKI quickly becomes infeasible in higher dimensions due to the $4^d$ entries per row of $W$ and curse of dimensionality for the number of points in a high-dimensional grid: specifically, $|U| = m^d$ for a grid with $m$ points in each dimension.

### 2.2 Sparse Grids

**Rectilinear grids.** We first give a formal construction of rectilinear grids, which will later be the foundation for sparse grids [7]. For a resolution index $l \in \mathbb{N}_0$, define the 1-d grid $\Omega_l$ as the centers

of $2^l$ equal partitions of the interval $[0, 1]$, which gives $\Omega_l := \{i/2^{l+1} \mid 1 \le i \le 2^{l+1} \text{ and } i \text{ is } \mathbf{odd}\}$. The fact that the position index $i$ must be odd implies that grids for any two different resolutions are *disjoint*.[2] Moreover, resolution-position index pairs $(l, i)$ uniquely specify grid points in the union $\bigcup_{l \in \mathbb{N}_0} \Omega_l$ of 1-d rectilinear grids.

To extend rectilinear grids to $d$ dimensions, let $\mathbf{l} \in \mathbb{N}_0^d$ denote a resolution vector. The corresponding rectilinear grid is given by $\Omega_\mathbf{l} := \otimes_{j=1}^d \Omega_{\mathbf{l}_j}$, where $\otimes$ denotes the Cartesian product. A grid point in $\Omega_\mathbf{l}$ is indexed by the pair $(\mathbf{l}, \mathbf{i})$ of a resolution vector $\mathbf{l}$ and *position vector* $\mathbf{i}$, where $(l_j, i_j)$ gives the position in the 1-d grid $\Omega_j$ for dimension $j$. This construction of rectilinear grids yields three essential properties that will facilitate formalizing sparse grids: (1) a grid $\Omega_\mathbf{l}$ is uniquely determined by its resolution vector $\mathbf{l}$, (2) grids $\Omega_\mathbf{l}$ and $\Omega_{\mathbf{l}'}$ with different resolution vectors are disjoint, (3) the size $|\Omega_\mathbf{l}| = 2^{\|\mathbf{l}\|_1}$ of a grid is determined by the $L_1$ norm of its resolution vector.

**Construction of Sparse Grids.** Sparse grids use rectilinear grids as their fundamental building block [22] and exploit the fact that resolution vectors uniquely identify different rectilinear grids. Larger grids are formed as the union of rectilinear grids with different resolution vectors. *Sparse* grids use all rectilinear grids with resolution vector having $L_1$ norm below a specified threshold. Formally, for a resolution index $\ell \in \mathbb{N}_0$, the sparse grid $\mathcal{G}_{\ell,d}$ in $d$ dimensions is $\mathcal{G}_{\ell,d} := \bigcup_{\mathbf{l}:\|\mathbf{l}\|_1 \le \ell} \Omega_\mathbf{l}$. Figure 1 illustrates the construction of the sparse grid $\mathcal{G}_{2,2}$ from smaller 2-d rectilinear grids with maximum resolution 2, i.e., $\{\Omega_\mathbf{l} \mid \|\mathbf{l}\|_1 \le 2\}$. The figure also illustrates another important fact: the sparse grid $\mathcal{G}_{\ell,d}$ has a total of $\Theta(2^\ell)$ distinct and equally spaced coordinates in each dimension, but many fewer total points than a dense $d$-fold Cartesian product of such 1-d grids, which would have $\Theta(2^{\ell d})$ points.

Sparse grids have a number of formal properties that are useful in algorithms and applications [22, 7]. Proposition 1 below summarizes the most relevant ones for our work. For completeness, a proof appears in appendix A. For more details, see Valentin [25].

---

**Proposition 1** (**Properties of Sparse Grid**). *Let $\mathcal{G}_{\ell,d} \subset [0,1]^d$ be a sparse grid with any resolution $\ell \in \mathbb{N}_0$ and dimension $d \in \mathbb{N}$. Then the following properties hold:*

(P1) $|\mathcal{G}_\ell^d| = \mathcal{O}(2^\ell \ell^{d-1})$,

(P2) $\forall \ell' \in \mathbb{N}, 0 \le \ell' \le \ell \implies \mathcal{G}_{\ell',d} \subseteq \mathcal{G}_{\ell,d}$,

(P3) $\mathcal{G}_{\ell,d} = \bigcup_{i=0}^\ell (\Omega_i \otimes \mathcal{G}_{\ell-i,d-1})$ *and* $\mathcal{G}_{\ell,1} = \bigcup_{i=0}^\ell \Omega_i$.

---

Property P1 shows that the size of sparse grid with $\Theta(2^\ell)$ points in each dimension grows more slowly than a dense grid with the same number of points in each dimension, since $\mathcal{O}(2^\ell \ell^{d-1}) \ll \mathcal{O}(2^{\ell d})$. Properties P2 and P3 are consequences of the structure of the set $\{\|\mathbf{l}\|_1 \le \ell\}$ and the sparse grid construction. Property P2 says that a sparse grid with smaller resolution is contained in one with higher resolution. Property P3 is a crucial property, and says that a $d$-dimensional sparse grid can be constructed via Cartesian products of 1-dimensional dense grids with sparse grids in $d-1$ dimensions.

## 3 Structured Kernel Interpolation on Sparse Grids

To scale kernel interpolation to higher dimensions, we propose to select inducing points $U = \mathcal{G}_{\ell,d}$ on a sparse grid and approximate the kernel matrix as $W K_{\mathcal{G}_{\ell,d}} W^T$ for a suitable interpolation matrix $W$ adapted to sparse grids. This will require fast matrix-vector multiplications with the sparse grid kernel matrix $K_{\mathcal{G}_{\ell,d}}$ and the interpolation matrix $W$. We show how to accomplish these two tasks in Sections 3.1 and 3.2 for the important case of stationary product kernels [10].

### 3.1 Fast Multiplication with the Sparse Grid Kernel Matrix

Algorithm 1 is an algorithm to compute $K_{\mathcal{G}_{\ell,d}} \mathbf{v}$ for any vector $\mathbf{v}$. The algorithm uses the following definitions. For any finite set $U$, let $K_U = [k(\mathbf{x}, \mathbf{x}')]_{\mathbf{x}, \mathbf{x}' \in U}$. The rows and columns of $K_U$ are "$U$-indexed", meaning the entries correspond to elements of $U$ under some arbitrary fixed ordering.

---

[2]Suppose $i/2^{l+1} = j/2^{k+1}$ are both grid points and $k > l$. Then $i = j2^{k-l}$ is even, a contradiction.

For $U \subseteq V$, we introduce a selection matrix $\mathcal{S}_{U,V}$ to map between $U$-indexed and $V$-indexed vectors. It has entries $(\mathcal{S}_{U,V})_{ij}$ equal to one if the $i$th element of $U$ is equal to the $j$th element of $V$, and zero otherwise. Also, $\mathcal{S}_{V,U} := \mathcal{S}_{U,V}^T$. For a $V$-indexed vector $\mathbf{z}_V$, the multiplication $\mathcal{S}_{U,V}\mathbf{z}_V$ produces a $U$-indexed vector by selecting entries corresponding to elements in $U$, and for a $U$-indexed vector $\mathbf{z}_U$, the multiplication $\mathcal{S}_{V,U}\mathbf{z}_U$ produces a $V$-indexed vector by inserting zeros for elements not in $U$.

---

**Algorithm 1** Sparse Grid Kernel-MVM Algorithm

---

**Input:** $\mathbf{v} \in \mathbb{R}^{|\mathcal{G}_{\ell,d}|}$ and $K_{\mathcal{G}_{\ell,d}} \in \mathbb{R}^{|\mathcal{G}_{\ell,d}| \times |\mathcal{G}_{\ell,d}|}$

**Output:** $\mathbf{u} = \mathbf{mvm}\left(K_{\mathcal{G}_{\ell,d}}, \mathbf{v}\right)$, where, $\mathbf{mvm}\left(K, \mathbf{v}\right)$ denotes $K\mathbf{v}$ obtained using this algorithm.

1: Let $V_i$ be the result of reshaping $\mathcal{S}_{\Omega_i \otimes \mathcal{G}_{\ell-i,d-1}, \mathcal{G}_{\ell,d}}\mathbf{v}$ into a $|\Omega_i| \times |\mathcal{G}_{\ell-i,d-1}|$ matrix; this contains entries of $\mathbf{v}$ corresponding to the $i$th grid in the decomposition of P3.

2: **if** $d = 1$ **then**

3:      return $\mathbf{u} = K_{\mathcal{G}_{\ell,1}}\mathbf{v}$                                          $\triangleright$ Base case

4: **end if**

5: $\triangleright$ Pre-computation

6: **for** $i = 0$ to $\ell$ **do**

7:      $\overline{A}_i = K_{\mathcal{G}_{i,1}}\mathcal{S}_{\mathcal{G}_{i,1},\Omega_i}V_i$

8:      $\overline{B}_i^T = \mathbf{mvm}\left(K_{\mathcal{G}_{\ell-i,d-1}}, V_i^T\right)$    $\triangleright$ Recursively multiply $K_{\mathcal{G}_{\ell-i,d-1}}$ with columns of $V_i^T$.

9: **end for**

10: $\triangleright$ Main loop

11: **for** $i = 0$ to $\ell$ **do**

12:      $A_i^T = \mathbf{mvm}\left(K_{\mathcal{G}_{\ell-i,d-1}}, \left(\sum_{j > i}\mathcal{S}_{\Omega_i,\mathcal{G}_{j,1}}\overline{A}_j\mathcal{S}_{\mathcal{G}_{\ell-j,d-1},\mathcal{G}_{\ell-i,d-1}}\right)^T\right)$   $\triangleright$ Recurse like line 8.

13:      $B_i = \mathcal{S}_{\Omega_i,\mathcal{G}_{i,1}}K_{\mathcal{G}_{i,1}}\left(\sum_{j \le i}\mathcal{S}_{\mathcal{G}_{i,1},\Omega_j}\overline{B}_j\mathcal{S}_{\mathcal{G}_{\ell-j,d-1},\mathcal{G}_{\ell-i,d-1}}\right)$

14:      $\mathbf{u}_i = \text{vec}\left[A_i\right] + \text{vec}\left[B_i\right]$

15: **end for**

---

**Theorem 1.** *Let $K_{\mathcal{G}_{\ell,d}}$ be the kernel matrix for a $d$-dimensional sparse grid with resolution $\ell$ for a stationary product kernel. For any $\mathbf{v} \in \mathbb{R}^{|\mathcal{G}_{\ell,d}|}$, Algorithm 1 computes $K_{\mathcal{G}_{\ell,d}}\mathbf{v}$ in $\mathcal{O}(\ell^d 2^\ell)$ time.*

The formal analysis and proof of Theorem 1 appears in Appendix B. The running time of $\mathcal{O}(\ell^d 2^\ell)$ is nearly linear in $|\mathcal{G}_{\ell,d}|$ and much faster asymptotically than the naive MVM algorithm that materializes the full matrix and has running time quadratic in $|\mathcal{G}_{\ell,d}|$.

Algorithm 1 is built on two high-level observations. First, the decomposition of Property P3 from Proposition 1 and the fact that the kernel follows product structure across dimensions are used to decompose the MVM into blocks, each of which is between sub-grids which are the product of a 1-dimensional rectilinear grid and a sparse grid in $d - 1$ dimensions. Therefore, the overall MVM computation can be recursively decomposed into MVMs with sparse grid kernel matrices in $d - 1$ dimensions. This observation is in part inspired by Zeiser [28], which also decomposes computation with the matrix on sparse grids by the resolution of first dimension. The base case occurs when $d = 1$. We assume that Toeplitz structure, which arises due to the kernel being stationary, is leveraged to perform this base case MVM in $O(\ell 2^\ell)$ time. Algorithm 1 can also be extended to *non-stationary* product kernels by using a standard MVM routine for the base case, which changes the overall running-time analysis but is still more efficient than the naive algorithm.

Secondly, by Property P2, the kernel matrix multiplication for any grid of resolution $\ell$ also includes the result of the multiplication for grids of lower resolution and the same number of dimensions. Thus, the results of the multiplications for many individual blocks can be obtained by using the appropriate selection operators with the result of the multiplication with the kernel matrix $K_{\mathcal{G}_{\ell-i,d-1}}$ in Line 12 of the algorithm. Further intuition and explanation are provided in the Appendix B.1.

**Improving batching efficiency.** The recursions in Lines 8 and 12 can be batched for efficiency, since both are multiplications with the same symmetric kernel matrix $K_{\mathcal{G}_{\ell,d}}$. Similarly, the recursion spawns many recursive multiplications with kernel matrices of the form $\mathcal{G}_{\ell',d'}$ for $0 \leq \ell' < \ell$ and $1 \leq d' < d$, and the calculation can be reorganized to batch all multiplications with each $K_{\mathcal{G}_{\ell',d'}}$. This is a significant savings, because there are only $d(\ell+1)$ distinct kernel matrices, but the recursion has a branching factor of $(\ell+1)$, so spawns many recursive calls with the same kernel matrices.

## 3.2 Sparse Interpolation For Sparse Grids

We now seek to construct the matrix $W$, which interpolates function values from the sparse grid $\mathcal{G}_{\ell,d}$ to training points $\mathbf{x}_i \in \mathbb{R}^d$, while ensuring that each row of $W$ is sparse enough to preserve efficiency of matrix-vector multiplications with $W$. This requires a sparse interpolation rule for sparse grids.

To set up the problem, we consider interpolating a function $f$ observed at points in a generic set $U$. Let $\mathbf{f}_U = (f(\mathbf{x}))_{u \in U}$. A linear interpolation rule for $U$ is a mapping $\mathbf{x} \mapsto \mathbf{w}_\mathbf{x} \in \mathbb{R}^{|U|}$ used to approximate $f(\mathbf{x})$ at an arbitrary point as $f(\mathbf{x}) \approx \mathbf{w}_\mathbf{x}^T \mathbf{f}_U$. The density of an interpolation rule is the maximum number of non-zeros in $\mathbf{w}_\mathbf{x}$ for any $\mathbf{x}$.

The *combination technique* for sparse grids constructs an interpolation rule by combining interpolation rules for the constituent rectilinear grids.

> **Proposition 2.** *For each* $\mathbf{l}$*, let* $\mathbf{w}_\mathbf{x}^\mathbf{l}$ *be an interpolation rule for the rectilinear grid* $\Omega_\mathbf{l}$ *with maximum density* $c$*. The combination technique gives an interpolation rule* $\mathbf{w}_\mathbf{x}$ *for the sparse grid* $\mathcal{G}_{\ell,d}$ *with density at most* $c \times \binom{\ell+d-1}{d-1}$*. The factor* $\binom{\ell+d-1}{d-1}$ *is the number of rectilinear grids included in* $\mathcal{G}_{\ell,d}$*.*

We use the combination technique to construct the sparse-grid interpolation coefficients $\mathbf{w}_{\mathbf{x}_i}$ for each training point $\mathbf{x}_i$ and stack them in the rows of $W$, which can be done in time proportional to the number of nonzeros. Details are given in Appendix A. The combination technique can use any base interpolation rule for the rectilinear grids, such as multilinear, cubic, or simplicial interpolation.

## 3.3 Simplicial Interpolation

The density of the interpolation rule is a critical consideration for kernel interpolation techniques—with or without sparse grids. Linear and cubic interpolation in $d$ dimensions have density $\Theta(2^d)$ and $\Theta(4^d)$, respectively. This "second curse of dimensionality" makes computations with $W$ intractable in higher dimensions independently of operations with the grid kernel matrix. For sparse grids, the density of $W$ increases by an additional factor of $\binom{\ell+d-1}{d-1}$.

We propose to use simplicial interpolation [11] for the underlying interpolation rule to avoid exponential growth of the density. Simplicial interpolation refers to a scheme where $\mathbb{R}^d$ is partitioned into simplices and a point $\mathbf{x}$ is interpolated using only the $d+1$ extreme points of the enclosing simplex, so the density of the interpolation rule is exactly $d+1$. Simplicial interpolation was previously proposed for sparse grid classifiers in [8]. In work closely related to ours, Kapoor et al. [13] used simplicial interpolation for GP kernel interpolation, with the key difference that they use the *permutohedral lattice* as the underlying grid, which has a number of nice properties but does not come equipped with fast specialized routines for kernel matrix multiplication.

In contrast to Kapoor et al. [13], we maintain rectilinear and/or sparse underlying grids, which preserve structure that enables fast kernel matrix multiplication. For rectilinear grids, this requires partitioning each hyper-rectangle into simplices, so the entire space is partitioned by simplices whose extreme points belong to the rectilinear grid. Then, within each simplex, the values at the extreme points are interpolated linearly. In general, there are different ways to partition hyper-rectangles into simplices—we use the specific scheme detailed in [11]. For sparse grids, we then use the combination technique, leading to overall density of $(d+1)\binom{\ell+d-1}{d-1}$. Table 1 provides the MVM complexities of $W$ and kernel matrices for different interpolation schemes and both grids. More details on how to perform simplicial interpolation with rectilinear and sparse grids are given in the Appendix B.2.

**Table 1:** The MVM complexities (i.e., $\mathrm{mvm}(\cdot)$) of the interpolation matrix $W$ and the kernel matrix $K_G$ for different interpolation bases and $d$-dimensional grids with $2^\ell$ unique points in each dimension and $n$ data points. A star indicates approaches proposed in this work. 'Dense' denotes a rectilinear grid.

| Grid | Basis | $\mathrm{mvm}(W)$ | $\mathrm{mvm}(K_G)$ |
|---|---|---|---|
| | Cubic | $\mathcal{O}(n \cdot 4^d)$ | $\mathcal{O}(\ell \cdot d \cdot 2^{\ell \cdot d})$ |
| Dense | Linear | $\mathcal{O}(n \cdot 2^d)$ | $\mathcal{O}(\ell \cdot d \cdot 2^{\ell \cdot d})$ |
| | Simplicial$^\star$ | $\mathcal{O}(n \cdot d^2)$ | $\mathcal{O}(\ell \cdot d \cdot 2^{\ell \cdot d})$ |
| | Cubic$^\star$ | $\mathcal{O}(n \cdot 4^d \cdot \binom{\ell+d-1}{d-1})$ | $\mathcal{O}(\ell^d \cdot 2^\ell)$ |
| Sparse | Linear$^\star$ | $\mathcal{O}(n \cdot 2^d \cdot \binom{\ell+d-1}{d-1})$ | $\mathcal{O}(\ell^d \cdot 2^\ell)$ |
| | Simplicial$^\star$ | $\mathcal{O}(n \cdot d^2 \cdot \binom{\ell+d-1}{d-1})$ | $\mathcal{O}(\ell^d \cdot 2^\ell)$ |

## 4 Experiments

In this section, we empirically evaluate the time and memory taken by Algorithm 1 for matrix-vector multiplication with the sparse grid kernel matrix, the accuracy of sparse grid interpolation and GP regression as the data dimension $d$ increases, and the accuracy of sparse grid kernel interpolation for GP regression on real higher-dimensional datasets from UCI. Hyper-parameters, data processing steps, and optimization details are given in Appendix C.1.

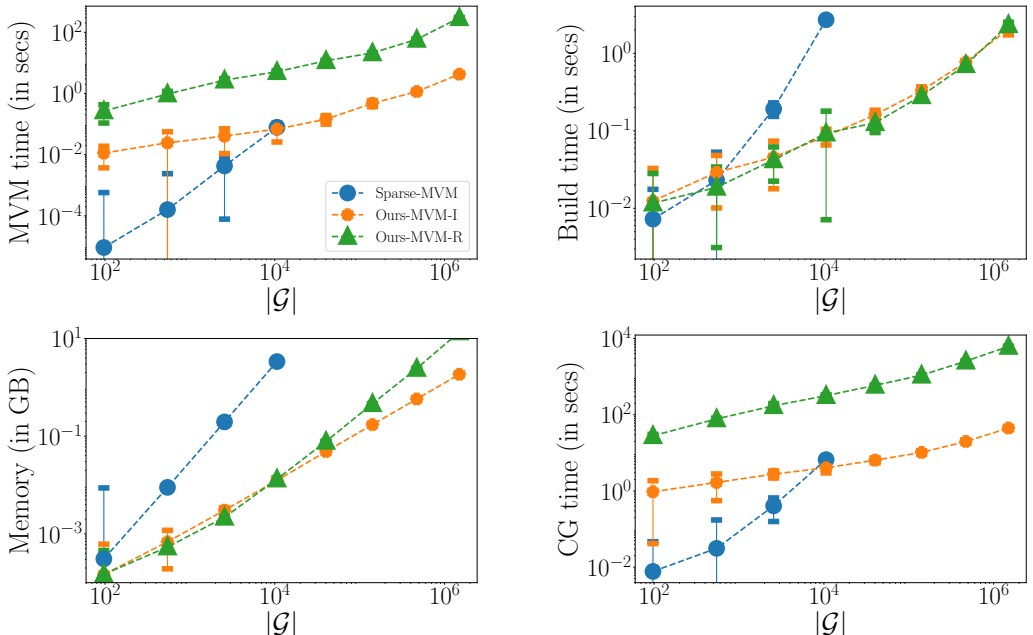

**Figure 2:** Matrix-vector multiplication (MVM) resource usage relative to sparse grid size for $d = 6$ and increasing resolution $\ell$. From top-left to bottom-right: time for one MVM operation, build (pre-processing) time for the kernel matrix, peak memory usage, and typical time to solve a linear system using CG (time for build plus 50 MVMs). Each plot shows the naive quadratic MVM algorithm (Sparse-MVM), the recursive implementation of Algorithm 1 (Ours-MVM-R), and the efficiently batched iterative implementation of Algorithm 1 (Ours-MVM-I). All measurements are averaged over 8 trials; error bars represent twice the standard error.

**Sparse grid kernel MVM complexity.** First, we evaluate the efficiency of MVM algorithms. We compare the basic and efficient implementations of Algorithm 1 to the naive algorithm, which constructs the full kernel matrix and scales quadratically with the sparse grid size. Algorithm 1 has a significant theoretical advantage in terms of both time and memory requirements as the grid size grows. Figure 2 illustrates this for $d = 6$ by comparing MVM time and memory requirements for sparse grids with resolutions $\ell \in \{2, \ldots, 9\}$ (roughly 100 to 1M grid points). The MVM time, preprocessing time, and memory consumption of Algorithm 1 all grow more slowly than the naive

algorithm, and the efficient implementation of Algorithm 1 is faster for $|\mathcal{G}|$ larger than about $10^4$, after which the naive algorithm also exceeds the 10 GB memory limit. For comparison, at $\ell = 6$ (about 40K grid points), Algorithm 1 uses only 0.05 GB of memory. These results indicate that the proposed algorithm is crucial for enabling sparse grid kernel interpolation in higher dimensions. Figure 2 also depicts the typical time to run GP inference (i.e., preprocessing time plus 50 MVM operations).

**Sparse grid interpolation and GP inference accuracy on synthetic data.** A significant advantage of sparse grids over dense rectilinear grids is their ability to perform accurate interpolation in higher dimensions. We demonstrate this for $d = 6$ by interpolating the function $f(\mathbf{x}) = \cos(\|\mathbf{x}\|_1)$ from observation locations on sparse grids of increasing resolution onto 200 random points sampled uniformly from $[0, 1]^d$. Figure 3, left, shows the interpolation error for dense and sparse grids with both cubic and simplicial interpolation. *Sparse (cubic)* is significantly more accurate than *Dense (cubic)*, and *Sparse (simplicial)* is more accurate than *Dense (simplicial)*.

We next evaluate the accuracy of GP inference in increasing dimensions. We keep the same function $f(\mathbf{x})$ and generate observations as $y = f(\mathbf{x}) + \mathcal{N}(0, 0.05)$. For all $d$, we use $\ell = 4$ for the sparse grid and compare to the dense grid with the closest possible total number of grid points (i.e., $\lceil \mathcal{G}_{4,d}^{1/d} \rceil$ points in each dimension)[3]. For $d \geq 8$, performance is better with sparse grids than dense grids for both interpolation schemes. Remarkably, our proposal to use simplicial interpolation with dense grids allows SKI to scale to $d = 10$, which is a significant improvement over prior work, in which SKI is typically infeasible for $d \geq 4$.

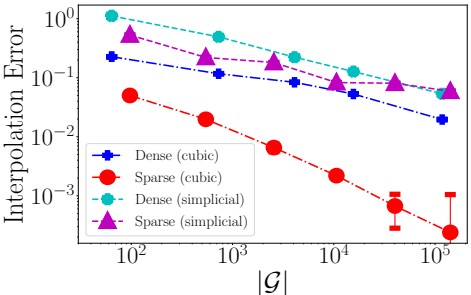
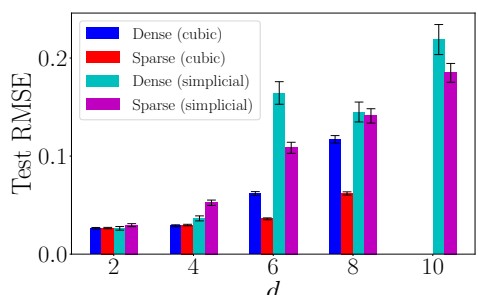

**Figure 3:** Comparing dense and sparse grids with cubic and simplicial interpolation schemes on synthetic data. Dense (cubic/simplicial) are dense grid methods with cubic/simplicial interpolation schemes; similarly, sparse (cubic/simplicial) are for sparse grids. Left: Function interpolation error vs. the grid size for $d = 6$. Right: Test root-mean-square error (RMSE) for GP regression for increasing dimensions. For both tasks, sparse grid methods outperform dense grids. For $d = 10$, both methods with cubic interpolation run out of GPU memory, which is 48 GB for this experiment.

**GP regression performance on UCI datasets.** To evaluate the effectiveness of our proposed methods for scaling GP kernel interpolation to higher dimensions, we consider all UCI [4] data sets with dimension $8 \leq d \leq 10$. We compare our proposed methods, Dense-grid (dense SKI with simplicial interpolation) and Sparse-grid (sparse-grid SKI with simplicial interpolation) to SGPR [24], SKIP [10], and Simplex-GP [13]. For SGPR, we report the best results using 256 or 512 inducing points. For SKIP, 100 points per dimension are used. For Simplex-GP [13], the blur stencil order is set to 1. Table 3 shows the root mean squared error (RMSE) for all methods. Our methods have performance comparable to and often better than SGPR and prior methods for scaling kernel interpolation to higher dimensions. This shows that sparse grids and simplicial interpolation can effectively scale SKI to higher dimensions to give a GP regression framework that is competitive with state-of-the-art approaches. We report additional results and analysis in Appendix C.

---

[3]We tried to match the sizes of both dense and sparse grids while ensuring that the dense grid always had at least as many points as the sparse grid, to give a fair comparison. Precisely, (d, dense grid size, sparse grid size) tuples are (2, 144, 129) , (4, 1296, 796), (6, 4096, 2561), (8, 6561, 6401), (10, 59049, 13441).

**Table 2:** Test root-mean-square-error (RMSE) on UCI regression datasets with dimensions (i.e., $8 \leq d \leq 10$). See text for algorithm descriptions and settings. All mean and standard deviations are computed over 3 trials. Pendulum is abbreviated as *Pendu* to accommodate table horizontally.

| Datasets ($d$) | SGPR | SKIP | Simplex-GP | Dense-grid | Sparse-grid |
|---|---|---|---|---|---|
| *Energy* (8) | $1.509 \pm 0.003$ | $5.762 \pm 0.000$ | $3.076 \pm 0.012$ | $1.333 \pm 0.009$ | $\mathbf{0.715} \pm 0.004$ |
| *Energy* (8) | $1.509 \pm 0.003$ | $5.762 \pm 0.000$ | $3.076 \pm 0.012$ | $1.333 \pm 0.009$ | $\mathbf{0.715} \pm 0.004$ |
| *Concrete* (8) | $12.727 \pm 0.018$ | $12.727 \pm 0.001$ | $12.727 \pm 0.000$ | $12.191 \pm 0.001$ | $\mathbf{8.655} \pm 0.002$ |
| *Kin40k* (8) | $\mathbf{0.168} \pm 0.009$ | $0.174 \pm 0.001$ | $0.287 \pm 0.003$ | $0.205 \pm 0.003$ | $0.483 \pm 0.000$ |
| *Fertility* (9) | $0.197 \pm 0.016$ | $0.183 \pm 0.000$ | $0.187 \pm 0.001$ | $\mathbf{0.182} \pm 0.000$ | $0.194 \pm 0.002$ |
| *Pendu* (9) | $\mathbf{1.948} \pm 0.021$ | $2.947 \pm 0.000$ | $2.577 \pm 0.009$ | $2.053 \pm 0.010$ | $2.103 \pm 0.015$ |
| *Protein* (9) | $0.605 \pm 0.001$ | $0.778 \pm 0.000$ | $\mathbf{0.582} \pm 0.018$ | $0.736 \pm 0.002$ | $0.595 \pm 0.001$ |
| *Solar* (10) | $0.790 \pm 0.026$ | $0.780 \pm 0.002$ | $0.792 \pm 0.000$ | $0.775 \pm 0.002$ | $\mathbf{0.748} \pm 0.006$ |

## 5   Related Works

Beyond SKI and its variants, a number scalable GP approximations have been investigated. Most notable are the different variants of sparse GP approximations [26, 23, 18]. For $m$ inducing points, these methods require either $\Omega(nm^2)$ time for direct solves or $\Omega(nm)$ time for approximate kernel MVMs in iterative solvers. While these methods generally do not leverage structured matrix algebra like the SKI framework, and thus have worse scaling in terms of the number of inducing points $m$, they may achieve comparable accuracy with smaller $m$, especially in higher dimensions. Some work utilizes the SKI framework to further boost the performance of sparse GP approximations [12].

A closely related work to ours is on improving the dimension scaling of SKI through the use of low-rank approximation and product structure [10]. In another very closely related work, Kapoor et al. [13] recently proposed to scale SKI to higher dimensions via interpolation on a permutohedral lattice. Like our work, they use simplicial interpolation. Unlike our work, the kernel matrix on the permutohedral lattice does not have special structure that admits fast exact multiplications; they instead use a locality-based approximation that takes into account the length scale of the kernel function by only considering pairs of grid points within a certain distance. Another related direction of research is the adaptation of sparse grid techniques for machine learning problems, e.g., classification and regression [16] and data mining [1]. These methods construct feature representations using sparse grid points [3], often by selecting a subset of grid points relevant for the dataset [6, 3].

## 6   Discussion

This work demonstrates that two classic numerical techniques, namely, sparse grids and simplicial interpolation, can be used to scale GP kernel interpolation to higher dimensions. SKI with sparse grids and simplicial interpolation has better or competitive regression accuracy compared to state-of-the-art GP regression approaches on several UCI benchmarking datasets with 8 to 10 dimensions.

**Limitations and future work.** Sparse grids and simplicial interpolation address two important bottlenecks when scaling kernel interpolation to higher dimensions. Sparse grids allow scalable matrix-vector multiplications with the grid kernel matrix, and simplicial interpolation allows scalable multiplications with the interpolation matrix $W$. The relatively large number of rectilinear grids used to form a sparse grid—i.e., the factor of $\binom{\ell+d-1}{d-1}$ in Proposition 2—is one limiting factor that makes multiplication by $W$ more costly. Future research could investigate methods to mitigate this extra cost, and explore the limits of scaling to even higher dimensions with sparse grids.

## Acknowledgments

This material is based upon work supported by the National Science Foundation under Grant Nos. 1661259, 1749854, 2046235 and 1763618, along with Adobe and Google Research Grants. We thank Javier Burroni, Abhinav Agarwal, Shiv Shankar and Shreyas Chaudhari for their feedback.

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
