# Supplementary Appendices

## A    Background – Omitted Details

### A.1    Sparse grids - Visualizations of grid points

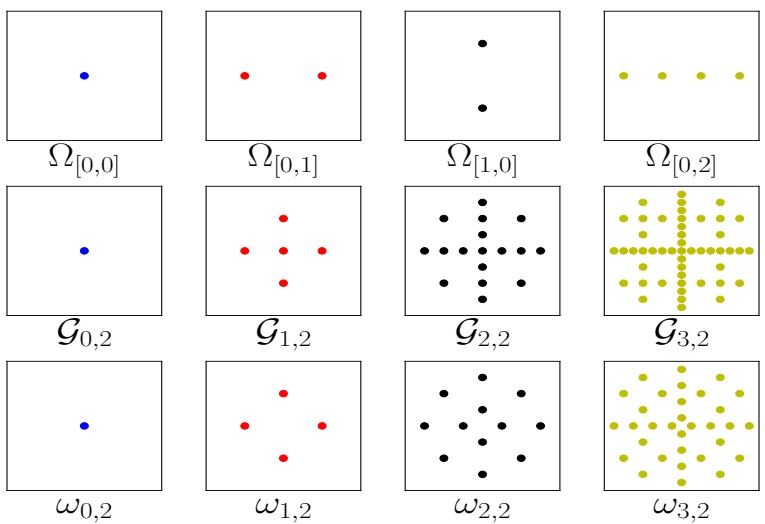

**Figure 4:** Visualizations of grid points on $[0,1] \times [0,1]$. From top to bottom: (row-1) rectilinear grids with different resolution vectors, (row-2) sparse grids with different resolutions for $d = 2$, and (row-3) incremental gain of grid points with resolution, i.e., $\omega_{\ell,d} := \mathcal{G}_{\ell,d} \setminus \bigcup_{0 \le l' \le \ell-1} \mathcal{G}_{l',d}$.

### A.2    Sparse grids – Properties and Hierarchical Interpolation

**Proposition 1 (Properties of Sparse Grid).** *Let $\mathcal{G}_{\ell,d} \subset [0,1]^d$ be a sparse grid with any resolution $\ell \in \mathbb{N}_0$ and dimension $d \in \mathbb{N}$. Then the following properties hold:*

(P1)  $|\mathcal{G}_\ell^d| = \mathcal{O}(2^\ell \ell^{d-1})$,

(P2)  $\forall \ell' \in \mathbb{N}, 0 \le \ell' \le \ell \implies \mathcal{G}_{\ell',d} \subseteq \mathcal{G}_{\ell,d}$,

(P3)  $\mathcal{G}_{\ell,d} = \bigcup_{i=0}^{\ell} \big( \Omega_i \otimes \mathcal{G}_{\ell-i,d-1} \big)$ and $\mathcal{G}_{\ell,1} = \bigcup_{i=0}^{\ell} \Omega_i$.

*Proof.*
(P1) – size of sparse grids:

$$
|\mathcal{G}_\ell^d| = \sum_{\{\mathbf{l} \in \mathbb{N}_0^d \,|\, \|\mathbf{l}\|_1 \le \ell\}} (2 \times 2^{\mathbf{l}_1 - 1})(2 \times 2^{\mathbf{l}_2 - 1}) \cdots (2 \times 2^{\mathbf{l}_d - 1})
$$

$$
= \sum_{\{\mathbf{l} \in \mathbb{N}_0^d \,|\, \|\mathbf{l}\|_1 = \ell\}} 2^{\|\mathbf{l}\|_1} \underbrace{\le}_{\text{summing over resolution}} \sum_{0 \le l' \le \ell} \underbrace{\binom{l' + d - 1}{d - 1}}_{\le l' + d - 1^{d-1}} 2^{l'}
$$

$$
\le (\ell + d - 1)^{d-1} \sum_{0 \le l' \le \ell} 2^{l'} = \mathcal{O}(\ell^{d-1} 2^\ell)
$$

(P2) – sparse grids with smaller resolution are in sparse grids with higher resolution:

$$
\forall \ell' \in \mathbb{N}, 0 \le \ell' \le \ell, \ \{\mathbf{l} \in \mathbb{N}_0^d \mid \|\mathbf{l}\|_1 \le \ell'\} \subseteq \{\mathbf{l} \in \mathbb{N}_0^d \mid \|\mathbf{l}\|_1 \le \ell\} \implies \mathcal{G}_{\ell',d} \subseteq \mathcal{G}_{\ell,d}
$$

(P3) – the recursive construction of sparse grids from rectilinear grids:

Let $\mathcal{L}_{\ell,d}$ be the set of $d$-dimensional vectors with $L_1$ norm bounded by $\ell$, i.e, $\mathcal{L}_{\ell,d} := \{\mathbf{l} \in \mathbb{N}_0^d \mid \|\mathbf{l}\|_1 \leq \ell\}$. Notice that $\mathcal{L}_{\ell,d}$ satisfies recursion similar to P3. I.e., $\mathcal{L}_{\ell,d} = \bigcup_{i=0}^{\ell}\big(\{i\}\otimes\mathcal{L}_{\ell-i,d-1}\big)$ and $\mathcal{L}_{\ell,1} = \bigcup_{i=0}^{\ell}\{i\}$. Next, P3 follows from the fact that $\mathcal{G}_{\ell,d} = \{\Omega_{\mathbf{l}} \mid \mathbf{l} \in \mathcal{L}_{\ell,d}\}$. $\qquad\square$

### A.2.1 Sparse grids - A hierarchical surplus linear interpolation approach

This subsection demonstrates how to use hierarchical surplus linear interpolation for kernel interpolation with sparse grids, however, we do not explore this method in our experiments. Nevertheless, we believe that the steps taken in adopting this method to kernel interpolation might of interest to readers and plausibly helpful for future exploration of kernel interpolation with sparse grids.

Our exposition to sparse grids in section A has been limited to specifying grid points, which can be extended to interpolation by associating basis functions with grid points.

We introduce $\mathbb{L}_{\ell,d} := \{(\mathbf{l},\mathbf{i}) \mid \mathbf{x}_{\mathbf{l},\mathbf{i}} \in \mathcal{G}_{\ell,d}\}$ and $\mathbf{x}_{\mathbf{l},\mathbf{i}} = \big[\mathbf{i}_1/2^{\mathbf{l}_1+1}, \cdots, \mathbf{i}_d/2^{\mathbf{l}_d+1}\big]$. Then, for any pair of resolution and index vectors $(\mathbf{l},\mathbf{i}) \in \mathbb{L}_{\ell,d}$, a tensorized hat function $\varphi_{\mathbf{l},\mathbf{i}}(\mathbf{x})$[4] is created such that it is centered at location of the grid-point corresponding to $(\mathbf{l},\mathbf{i})$ and has support for a symmetric interval of length $2^{-\mathbf{l}_i}$ in the $i$ dimension. For any function $f : \mathbb{R}^d \mapsto R$, the sparse grid *interpolant* rule $f^{\ell} : \mathbb{R}^d \mapsto R$ is given as:

$$f^{\ell}(\mathbf{x}) = \sum_{(\mathbf{l},\mathbf{i})\in\mathbb{L}_{\ell,d}} \sum_{\boldsymbol{\delta}\in\Delta^d} (-2)^{-\|\boldsymbol{\delta}\|_0} f(\mathbf{x}_{\mathbf{l},\mathbf{i}+\boldsymbol{\delta}})\varphi_{\mathbf{l},\mathbf{i}}(\mathbf{x}) \tag{1}$$

where $\Delta^d = \{-1,\ 0,\ 1\}^d$ is stencil evaluation of the function $f$ centered at grid-point $\mathbf{x}_{\mathbf{l},\mathbf{i}}$ [16]. Concretely, $\mathbf{x}_{\mathbf{l},\mathbf{i}+\boldsymbol{\delta}} \in \mathbb{R}^d$ has $k^{th}$ position equal to $(\mathbf{i}_k + \boldsymbol{\delta}_k)2^{-\mathbf{l}_k}$. Figure 5 provides an illustration of the above sparse grid *interpolant* rule for simple 1-dimensional functions. Notice that it progressively gets more accurate as the resolution level $\ell$ increases.

Next to interpolate the kernel function, we need $f^{\ell}(\mathbf{x}) = \mathbf{w}_{\mathbf{x}}\theta$, where $\mathbf{w}_{\mathbf{x}} \in \mathbb{R}^{1\times|G_{\ell}^d|}$ and $\theta \in \mathbb{R}^{|G_{\ell}^d|}$ is the evaluation of $f$ on $\mathcal{G}_{\ell,d}$. As given such a formula, we can write $k(\mathbf{x},\mathbf{x}') \approx \text{cov}(f^{\ell}(\mathbf{x}), f^{\ell}(\mathbf{x}')) = \mathbf{w}_{\mathbf{x}}K_{\mathcal{G}_{\ell,d}}\mathbf{w}_{\mathbf{x}}^T$, where $K_{\mathcal{G}_{\ell,d}}$ is the true kernel matrix on sparse grid. Furthermore, by stacking interpolation weights $\mathbf{w}_{\mathbf{x}}$ into $W$ matrix for all data points similar to SKI, we can approximate the kernel matrix as $\tilde{K}_X = WK_{\mathcal{G}_{\ell,d}}W^T$.

---

**Claim 1.** $\forall \mathbf{x} \in \mathbb{R}^d, \ell \in \mathbb{N}_0, \exists \mathbf{w}_{\mathbf{x}}$ *s.t.* $f^{\ell}(\mathbf{x}) = \mathbf{w}_{\mathbf{x}}\theta$ *where* $\theta \in \mathbb{R}^{|\mathcal{G}^{\ell,d}|}$ *is the evaluation of $f$ on $\mathcal{G}_{\ell,d}$, i.e, $\theta$ is made of $\{f(\mathbf{x}_{\mathbf{l},\mathbf{i}}) \mid \mathbf{x}_{\mathbf{l},\mathbf{i}} \in \mathcal{G}_{\ell,d}\}$ and indexed to match the columns of $\mathbf{w}_{\mathbf{x}}\theta$.*

---

*Proof.* For brevity, we introduce, $Q_{\ell,d} := \mathbb{L}_{\ell,d} \setminus \bigcup_{0\leq l'\leq\ell-1} \mathbb{L}_{l',d}$, which is a partition of $\mathbb{L}_{\ell,d}$ based on the resolution of grid points, i.e., $\mathbb{L}_{\ell,d} = \bigcup_{0\leq l'\leq\ell} Q_{l',d}$.

---

[4]$\varphi_{\mathbf{l},\mathbf{i}}(\mathbf{x}) := \prod_{k=1}^d \varphi_{\mathbf{l}_k,\mathbf{i}_k}(\mathbf{x}_k)$ where $\forall k, \varphi_{\mathbf{l}_k,\mathbf{i}_k}(\mathbf{x}_k) = \varphi(\frac{\mathbf{x}_k-\mathbf{i}_k\cdot2^{-\mathbf{l}_k}}{2^{-\mathbf{l}_k}})$ and $\varphi(x) := \max\{1-|x|, 0\}$.

$$f^{\ell}(\mathbf{x}) = \sum_{(\mathbf{l},\mathbf{i})\in\mathbb{L}_{\ell,d}} \sum_{\boldsymbol{\delta}\in\Delta^d} (-2)^{-\|\boldsymbol{\delta}\|_0} f(\mathbf{x}_{\mathbf{l},\mathbf{i}+\boldsymbol{\delta}})\varphi_{\mathbf{l},\mathbf{i}}(\mathbf{x})$$

$$= \sum_{0\le l'\le \ell} \sum_{(\mathbf{l},\mathbf{i})\in Q_{l',d}} \left( \sum_{\boldsymbol{\delta}\in\{-1\ 0\ 1\}^d} 2^{-\|\boldsymbol{\delta}\|_0} f(\mathbf{x}_{\mathbf{l},\mathbf{i}+\boldsymbol{\delta}}) \right) \varphi_{\mathbf{l},\mathbf{i}}(\mathbf{x})$$

$$= \sum_{0\le l'\le \ell} \sum_{(\mathbf{l},\mathbf{i})\in Q_{l',d}} \left( \sum_{\boldsymbol{\delta}\in\{-1\ 0\ 1\}^d} 2^{-\|\boldsymbol{\delta}\|_0} f(\mathbf{x}_{\mathbf{l},\mathbf{i}+\boldsymbol{\delta}}) \right) \varphi_{\mathbf{l},\mathbf{i}}(\mathbf{x})$$

$$= \sum_{0\le l'\le \ell} \sum_{(\mathbf{l},\mathbf{i}'-\boldsymbol{\delta})\in Q_{l',d}} \sum_{\boldsymbol{\delta}\in\{-1\ 0\ 1\}^d} 2^{-\|\boldsymbol{\delta}\|_0} \varphi_{\mathbf{l},\mathbf{i}'-\boldsymbol{\delta}}(\mathbf{x})f(\mathbf{x}_{\mathbf{l},\mathbf{i}'}) \quad \text{by substitution } \mathbf{i} = \mathbf{i}' - \boldsymbol{\delta}$$

$$= \sum_{0\le l'\le \ell} \left( \sum_{(\mathbf{l},\mathbf{i}'-\boldsymbol{\delta})\in Q_{l',d}} \sum_{\boldsymbol{\delta}\in\{-1\ 0\ 1\}^d} 2^{-\|\boldsymbol{\delta}\|_0} \varphi_{\mathbf{l},\mathbf{i}'-\boldsymbol{\delta}}(\mathbf{x}) \right) f(\mathbf{x}_{\mathbf{l},\mathbf{i}'})$$

$$= \sum_{0\le l'\le \ell} \left( \sum_{(\mathbf{l},\mathbf{i}')\in Q_{l',d}} \sum_{\boldsymbol{\delta}\in\{-1\ 0\ 1\}^d} 2^{-\|\boldsymbol{\delta}\|_0} \varphi_{\mathbf{l},\mathbf{i}'+\boldsymbol{\delta}}(\mathbf{x}) \right) f(\mathbf{x}_{\mathbf{l},\mathbf{i}'})$$

$$= \sum_{0\le l'\le \ell} \sum_{(\mathbf{l},\mathbf{i}')\in Q_{l',d}} \sum_{\boldsymbol{\delta}\in\{-1\ 0\ 1\}^d} 2^{-\|\boldsymbol{\delta}\|_0} \varphi_{\mathbf{l},\mathbf{i}'+\boldsymbol{\delta}}(\mathbf{x})f(\mathbf{x}_{\mathbf{l},\mathbf{i}'})$$

$$= \sum_{0\le l'\le \ell} \sum_{(\mathbf{l},\mathbf{i})\in Q_{l',d}} \sum_{\boldsymbol{\delta}\in\{-1\ 0\ 1\}^d} 2^{-\|\boldsymbol{\delta}\|_0} \varphi_{\mathbf{l},\mathbf{i}+\boldsymbol{\delta}}(\mathbf{x})f(\mathbf{x}_{\mathbf{l},\mathbf{i}})$$

$$= \sum_{(\mathbf{l},\mathbf{i})\in\mathbb{L}_{\ell,d}} \sum_{\boldsymbol{\delta}\in\{-1\ 0\ 1\}^d} 2^{-\|\boldsymbol{\delta}\|_0} \varphi_{\mathbf{l},\mathbf{i}+\boldsymbol{\delta}}(\mathbf{x})f(\mathbf{x}_{\mathbf{l},\mathbf{i}})$$

Though it may seem that not all $\varphi_{\mathbf{l},\mathbf{i}+\boldsymbol{\delta}}$ are on sparse grid as the components of $\mathbf{i} + \boldsymbol{\delta}$ can be even. Fortunately, it is true as $x_{l,i} = x_{l+1,2i}$ by the construction of sparse grids, and we can apply the following transformation to uniquely project $(\mathbf{l}, \mathbf{i} + \boldsymbol{\delta})$ on $\mathcal{G}_{\ell,d}$ as follows:

$$\Delta(\mathbf{l}, \mathbf{i}, \boldsymbol{\delta}) = \left( \mathbf{l} - \#^2(\mathbf{i}+\boldsymbol{\delta}), \frac{\mathbf{i}}{2^{\#^2(\mathbf{i}+\boldsymbol{\delta})}} \right),$$

where $\#^2$ computes exponent of 2 in its prime factorization component-wise. Notice that output of $\Delta(\mathbf{l}, \mathbf{i}, \boldsymbol{\delta})$ is bound to be in $\mathcal{G}_{\ell,d}$ as the resultant-position index pair (i.e., $(\mathbf{l}, \mathbf{i})$) will have level index $l \le \ell$ and position index $i$ to be odd, for all components, respectively.

$$f^{\ell}(\mathbf{x}) = \sum_{(\mathbf{l},\mathbf{i})\in\mathbb{L}_{\ell,d}} \underbrace{\sum_{\boldsymbol{\delta}\in\{-1\ 0\ 1\}^d} (-2)^{-\|\boldsymbol{\delta}\|_0} \varphi_{\Delta(\mathbf{l},\mathbf{i},\boldsymbol{\delta})}(\mathbf{x})}_{:=\mathbf{w}_{\mathbf{l},\mathbf{i}}(\mathbf{x})} f(\mathbf{x}_{\mathbf{l},\mathbf{i}}) \tag{2}$$

By setting $\mathbf{w}_{\mathbf{x}}$ as $\mathbf{w}_{\mathbf{l},\mathbf{i}}(\mathbf{x})$ from above equation for all $(\mathbf{l}, \mathbf{i}) \in \mathbb{L}_{\ell,d}$, we have $f^{\ell}(\mathbf{x}) = \mathbf{w}_{\mathbf{x}}\theta$.

$\square$

## A.3 Combination Technique for Sparse Grid Interpolation

The combination technique [15] provides yet another *interpolant* rule for sparse grids. Formally, for any function $f : \mathbb{R}^d \mapsto R$, the sparse grid combination technique *interpolant* rule $f_c^{\ell} : \mathbb{R}^d \mapsto R$ is:

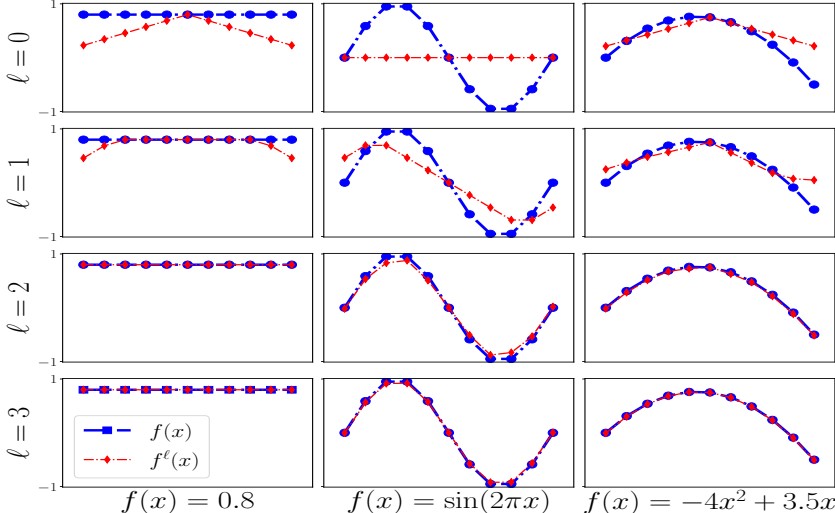

**Figure 5:** Interpolation on sparse grids with increasing resolution (i.e., $\ell$) for 1-dimensional functions.

$$f_c^\ell(\mathbf{x}) = \sum_{q=0}^{d-1} (-1)^q \binom{d-1}{q} \sum_{\{\mathbf{l} \in \mathbb{N}_0^d \,|\, \|\mathbf{l}\|_1 = \ell - q\}} f_{\Omega_\mathbf{l}}(\mathbf{x}), \tag{3}$$

where, $f_{\Omega_\mathbf{l}}$ is an *interpolant* rule on the rectilinear grid $\Omega_\mathbf{l}$ [25, 15]. Consequently, it is trivial to extend simplicial and cubic interpolation from rectilinear grid to sparse grids. Notice that the interpolant $f_c^\ell$ uses a smaller set of grids, i.e., only $\{\Omega_\mathbf{l} \mid \max\{\ell - d, 0\} < \|\mathbf{l}\|_1 \le \ell\}$. Nevertheless, $\forall\ \ell\ \&\ d$, the rectilinear grids used in $f_c^\ell$ are always contained in $\mathcal{G}_{\ell,d}$.

Equation 3 prototypes the construction of interpolation weights $W$ matrix. Concretely, for each dense grid $\Omega_\mathbf{l}$ in the sparse grid, interpolation weights are computed and stacked along columns. After that columns are scaled by factor $(-1)^q \binom{d-1}{q}$ to satisfy $f_c^\ell$.

# B  Structured Kernel Interpolation on Sparse Grids – Omitted Details

## B.1  Fast Multiplication with Sparse Grid Kernel Matrix

**Indexing the kernel matrix** $K_{\mathcal{G}_{\ell,d}}$. Recall P3 from proposition 1, i.e., the recursive construction of sparse grids, $\mathcal{G}_{\ell,d} = \bigcup_{i=0}^{\ell} \left( \Omega_i \otimes \mathcal{G}_{\ell-i,d-1} \right)$. We say $\mathcal{G}_{\ell,d}^i := \Omega_i \otimes \mathcal{G}_{\ell-i,d-1}$. From P3, we know that $K_{\mathcal{G}_{\ell,d}}$ can be written as the block matrix such that both rows and columns are indexed by all $\mathcal{G}_{\ell,d}^i$. For all combinations of combinations $\mathcal{G}_{\ell,d}^i$, $the K_{\mathcal{G}_{\ell,d}}$ is a $(\ell+1) \times (\ell+1)$ block matrix.

Similarly, without the loss of generality, any arbitrary vector $\mathbf{v} \in \mathbb{R}^{|\mathcal{G}_{\ell,d}|}$ is indexed using $\mathcal{G}_{\ell,d}^i$ and the output vector after an MVM operation also follows indexing by $\mathcal{G}_{\ell,d}^i$. Concretely, we let $\mathbf{u} = K_{\mathcal{G}_{\ell,d}} \mathbf{v}$, then $\forall\ 0 \le i \le \ell$, we can write $\mathbf{u}_i = \sum_{j=1}^{\ell} \tilde{\mathbf{v}}_{ij}$, where $\tilde{\mathbf{v}}_{ij} = K_{\mathcal{G}_{\ell,d}^i, \mathcal{G}_{\ell,d}^j} \mathbf{v}_j$, rows of $\mathbf{v}$, rows of $\mathbf{u}$ and rows and columns of $K$, are indexed by the $\mathcal{G}_{\ell,d}^i$.

**Structure and redundancy in the kernel sub-matrices**. Note that $\mathcal{G}_{\ell,d}^i$ is the Cartesian product between rectilinear grid $\Omega_i$ and sparse grid $\mathcal{G}_{\ell-i,d-1}$ imposing Kronecker structure on the matrix $K_{\mathcal{G}_{\ell,d}^i, \mathcal{G}_{\ell,d}^j}$, given that $k$ is a product kernel. As a result, for each $\mathbf{u}_i$, we have:

$$\tilde{\mathbf{v}}_{ij} = \text{vec} \left[ K_{\Omega_i, \Omega_j} \, \text{mat}(\mathbf{v}_j) K_{\mathcal{G}_{\ell-i,d-1}, \mathcal{G}_{\ell-j,d-1}}^T \right] \tag{4}$$

where $\text{vec}$ and $\text{mat}$ are standard matrix reshaping operators used in multiplying vectors with a Kronecker product of two matrices. Observe that both $K_{\Omega_i, \Omega_j}$ and $K_{\mathcal{G}_{\ell-i,d-1}, \mathcal{G}_{\ell-j,d-1}}$ are rectangular

and have many common entries across different pairs of $i$ and $j$. For instance, $\forall j > i$, $K_{\Omega_i,\Omega_j} \subseteq K_{\mathcal{G}_{j,1}}$, similarly, we also have, $\forall j \leq i$, $K_{\mathcal{G}_{\ell-i,d-1},\mathcal{G}_{\ell-j,d-1}} \subseteq K_{\mathcal{G}_{\ell-i,d-1}}$ as $\mathcal{G}_{\ell-j,d-1} \subseteq \mathcal{G}_{\ell-i,d-1}$.

**Efficient ordering for Kronecker product and exploiting redundancy in kernel sub-matrices**. Next, observing the two orders of computation for Equation 4 (i.e., first multiplying $K_{\Omega_i,\Omega_j}$ with versus $K_{\mathcal{G}_{\ell-i,d-1},\mathcal{G}_{\ell-j,d-1}}^T$) and to leverage the above-mentioned redundancy, we divide the computation as below[5]:

$$\mathbf{u}_i = \mathbf{a}_i + \mathbf{b}_i, \quad \text{where,} \quad \mathbf{a}_i = \sum_{j>i}^{\ell} \tilde{\mathbf{v}}_{ij}, \quad \text{and} \quad \mathbf{b}_i = \sum_{j=0}^{i} \tilde{\mathbf{v}}_{ij}. \tag{5}$$

In Algorithm 1, $A_i$ and $B_i$ are such that $\text{vec}(A_i) = \mathbf{a}_i$ and $\text{vec}(A_i) = \mathbf{b}_i$.

---

**Claim 2.** *With $\overline{A}_i := K_{\mathcal{G}_{i,1}} \mathcal{S}_{\mathcal{G}_{i,1},\Omega_i} \text{mat}(\mathbf{v})$, $\forall 0 \leq i \leq \ell$, $\mathbf{a}_i$ can be given as follows:*

$$\mathbf{a}_i = \text{vec}\left[ \left( \sum_{j>i} \mathcal{S}_{\Omega_i,\mathcal{G}_{j,1}} \overline{A}_j \mathcal{S}_{\mathcal{G}_{\ell-j,d-1},\mathcal{G}_{\ell-i,d-1}} \right) K_{\mathcal{G}_{\ell-i,d-1}} \right]$$

---

*Proof.*

$$\mathbf{a}_i = \sum_{j>i}^{\ell} \tilde{\mathbf{v}}_{ij}$$

$$= \sum_{j>i}^{\ell} \text{vec}\left[ K_{\Omega_i,\Omega_j} \text{mat}(\mathbf{v}_j) K_{\mathcal{G}_{\ell-i,d-1},\mathcal{G}_{\ell-j,d-1}}^T \right] \text{ (from the Equation 4)}$$

$$= \sum_{j>i}^{\ell} \text{vec}\left[ K_{\Omega_i,\Omega_j} \text{mat}(\mathbf{v}_j) \mathcal{S}_{\mathcal{G}_{\ell-j,d-1},\mathcal{G}_{\ell-i,d-1}} K_{\mathcal{G}_{\ell-i,d-1}} \right] \text{ (by expanding the kernel matrix)}$$

$$= \text{vec}\left[ \sum_{j>i}^{\ell} \left( K_{\Omega_i,\Omega_j} \text{mat}(\mathbf{v}_j) \mathcal{S}_{\mathcal{G}_{\ell-j,d-1},\mathcal{G}_{\ell-i,d-1}} \right) K_{\mathcal{G}_{\ell-i,d-1}} \right] \text{ (using linearity of operations)}$$

$$= \text{vec}\left[ \sum_{j>i}^{\ell} \left( K_{\Omega_i,\mathcal{G}_{j,1}} \mathcal{S}_{\mathcal{G}_{j,1},\Omega_j} \text{mat}(\mathbf{v}) \mathcal{S}_{\mathcal{G}_{\ell-j,d-1},\mathcal{G}_{\ell-i,d-1}} \right) K_{\mathcal{G}_{\ell-i,d-1}} \right] \text{ (by expanding kernel matrix)}$$

$$= \text{vec}\left[ \sum_{j>i}^{\ell} \left( \mathcal{S}_{\Omega_i,\mathcal{G}_{j,1}} K_{\mathcal{G}_{j,1}} \mathcal{S}_{\mathcal{G}_{j,1},\Omega_j} \text{mat}(\mathbf{v}) \mathcal{S}_{\mathcal{G}_{\ell-j,d-1},\mathcal{G}_{\ell-i,d-1}} \right) K_{\mathcal{G}_{\ell-i,d-1}} \right]$$

$$= \text{vec}\left[ \sum_{j>i}^{\ell} \left( \mathcal{S}_{\Omega_i,\mathcal{G}_{j,1}} \overline{A}_j \mathcal{S}_{\mathcal{G}_{\ell-j,d-1},\mathcal{G}_{\ell-i,d-1}} \right) K_{\mathcal{G}_{\ell-i,d-1}} \right] \text{ (enables pre-computation using } \overline{A}_j\text{)}$$

$\square$

Intuitively, the claim 2 demonstrates rationale behind the operator $\mathcal{S}$, because (1) it reduces the computation from $\frac{\ell^2}{2}$ MVM with $K_{\Omega_i,\Omega_j}$ to only $\ell$ MVM with $K_{\mathcal{G}_{i,1}}$ Toeplitz matrices via pre-computing $\overline{A}_i$, and (2) it requires only $\ell$ MVM with $K_{\mathcal{G}_{\ell-i,d-1}}$ instead of $\frac{\ell^2}{2}$ MVMs with $K_{\mathcal{G}_{\ell-i,d-1},\mathcal{G}_{\ell-j,d-1}}$ via exploiting linearity of operations involved. Following analogous steps, $\mathbf{b}_i$ can be derived $\mathbf{b}_i = \text{vec}(B_i)$.

---

[5]This is in part inspired by Zeiser [28] as our algorithm also orders computation by first dimension of sparse grid (i.e., $\mathcal{G}_{\ell,d}^i$).

> **Claim 3.** *With $\overline{B}_i := V_i K_{\mathcal{G}_{\ell-i,d-1}}$, $\forall 0 \leq i \leq \ell$, $\mathbf{b}_i$ can be given as follows:*
>
> $$\mathbf{b}_i = \text{vec}\left[\mathcal{S}_{\Omega_i,\mathcal{G}_{i,1}} K_{\mathcal{G}_{i,1}}\left(\sum_{j\leq i}\mathcal{S}_{\mathcal{G}_{i,1},\Omega_j}\overline{B}_j \mathcal{S}_{\mathcal{G}_{\ell-j,d-1},\mathcal{G}_{\ell-i,d-1}}\right)\right]$$

*Proof.*

$$\begin{aligned}
\mathbf{b}_i &= \sum_{j\leq i}\tilde{\mathbf{v}}_{ij}\\
&= \sum_{j\leq i}\text{vec}\left[K_{\Omega_i,\Omega_j}\,\text{mat}(\mathbf{v}_j)K_{\mathcal{G}_{\ell-i,d-1},\mathcal{G}_{\ell-j,d-1}}^T\right]\text{ (from Equation 4)}\\
&= \sum_{j\leq i}\text{vec}\left[K_{\Omega_i,\Omega_j}V_j K_{\mathcal{G}_{\ell-j,d-1}}\mathcal{S}_{\mathcal{G}_{\ell-j,d-1},\mathcal{G}_{\ell-i,d-1}}\right]\\
&= \sum_{j\leq i}\text{vec}\left[\mathcal{S}_{\Omega_i,\mathcal{G}_{i,1}}K_{\mathcal{G}_{i,1},\Omega_j}V_j K_{\mathcal{G}_{\ell-j,d-1}}\mathcal{S}_{\mathcal{G}_{\ell-j,d-1},\mathcal{G}_{\ell-i,d-1}}\right]\\
&= \sum_{j\leq i}\text{vec}\left[\mathcal{S}_{\Omega_i,\mathcal{G}_{i,1}}K_{\mathcal{G}_{i,1}}\mathcal{S}_{\mathcal{G}_{i,1},\Omega_j}V_j K_{\mathcal{G}_{\ell-j,d-1}}\mathcal{S}_{\mathcal{G}_{\ell-j,d-1},\mathcal{G}_{\ell-i,d-1}}\right]\\
&= \text{vec}\left[\mathcal{S}_{\Omega_i,\mathcal{G}_{i,1}}K_{\mathcal{G}_{i,1}}\left(\sum_{j\leq i}\mathcal{S}_{\mathcal{G}_{i,1},\Omega_j}\overline{B}_j\mathcal{S}_{\mathcal{G}_{\ell-j,d-1},\mathcal{G}_{\ell-i,d-1}}\right)\right]
\end{aligned}$$

$\square$

> **Theorem 1.** *Let $K_{\mathcal{G}_{\ell,d}}$ be the kernel matrix for a $d$-dimensional sparse grid with resolution $\ell$ for a stationary product kernel. For any $\mathbf{v} \in \mathbb{R}^{|\mathcal{G}_{\ell,d}|}$, Algorithm 1 computes $K_{\mathcal{G}_{\ell,d}}\mathbf{v}$ in $\mathcal{O}(\ell^d 2^\ell)$ time.*

*Proof.* **The correctness of the Algorithm 1**. Equation 5, claim 2 and claim 3 establish the correctness of the output of Algorithm 1, i.e., it computes $K_{\mathcal{G}_{\ell,d}}\mathbf{v}$.

**On the complexity of Algorithm 1**. (We prove it by induction on $d$.)

**Base case:** For any $\ell$ and $d = 1$, the algorithm utilizes Toeplitz multiplication which require only $|\mathcal{G}_{\ell,1}|\log|\mathcal{G}_{\ell,1}|$, as $|\mathcal{G}_{\ell,1}| = 2^{\ell+1}$, total required computation is $\mathcal{O}(\ell 2^\ell)$.

**Inductive step:** We assume that the complexity holds, i.e., $\mathcal{O}(\ell^d 2^\ell)$ for $d-1$, then it's sufficient to show that Algorithm 1 needs only $\mathcal{O}(\ell^d 2^\ell)$ for $d$, in order to complete the proof. Below, we establish the same separately for both pre-computation steps (i.e., Line 6 to 9) and the main loop (i.e., Line 11 to 15) of Algorithm 1. Before that, we state an important fact for the analysis of remaining steps:

$$\sum_{i=0}^{\ell}|\mathcal{G}_{i,1}|\times|\mathcal{G}_{\ell-i,d-1}| = \sum_{i=0}^{\ell}2\times|\Omega_i|\times|\mathcal{G}_{\ell-i,d-1}|\underbrace{=}_{P3}2|\mathcal{G}_{\ell,d}| = O(\ell^{d-1}2^\ell) \quad (6)$$

**Analysis of the pre-computation steps**.

- For reshaping $\mathbf{v}$ into $V_i$'s, we need $\sum_{i=0}^{\ell}|\Omega_i|\times|\mathcal{G}_{\ell-i,d-1}| = |\mathcal{G}_{\ell,d}| = O(\ell^{d-1}2^\ell)$.

- For the rearrangement $V_i$ into $\mathcal{S}_{\mathcal{G}_{i,1},\Omega_i}V_i$, we need $\sum_{i=0}^{\ell}|\mathcal{G}_{i,1}|\times|\mathcal{G}_{\ell-i,d-1}|$ as operator $\mathcal{S}$ maps $V_i$ directly into the result. Therefore, we need $O(\ell^{d-1}2^\ell)$ using Equation 6.

- For the $\overline{A}_i$ step:

  - $\forall i$, $|\mathcal{G}_{\ell-i,d-1}|$ vectors are multiplied with Toeplitz matrix of size $|\mathcal{G}_{i,1}|\times|\mathcal{G}_{i,1}|$,

- so total computation for line 7 is, $\sum_{i=0}^{\ell} |\mathcal{G}_{i,1}| \log |\mathcal{G}_{i,1}| \times |\mathcal{G}_{\ell-i,d-1}| = \sum_{i=0}^{\ell} i \times |\mathcal{G}_{i,1}| \times |\mathcal{G}_{\ell-i,d-1}| \leq \sum_{i=0}^{\ell} \ell \times |\mathcal{G}_{i,1}| \times |\mathcal{G}_{\ell-i,d-1}| \underbrace{\leq}_{\text{Eq. 6}} \ell \times \mathcal{O}(\ell^{d-1}2^{\ell}) = \mathcal{O}(\ell^d 2^{\ell})$.

- For the $\overline{B}_i$ step:
    - $\forall i, |\Omega_i|$ vectors need to be multiplied with $K_{\mathcal{G}_{\ell-i,d-1}}$,
    - using induction, total computation for line 8 is, $\sum_{i=0}^{\ell} 2^i \times (\ell-i)^d 2^{\ell-i} = 2^{\ell} \sum_{i=0}^{\ell} (\ell - i)^d \leq 2^{\ell} \sum_{i=0}^{\ell} \ell^{d-1} = \mathcal{O}(\ell^d 2^{\ell})$.

**Analysis of the main loop**.

- For the $A_i$ step,
    - $\ell - i$ rearrangements and summations (i.e., $\sum_{j>i} \mathcal{S}_{\Omega_i,\mathcal{G}_{j,1}} \overline{A}_j \mathcal{S}_{\mathcal{G}_{\ell-j,d-1},\mathcal{G}_{\ell-i,d-1}}$) are performed simultaneously (i.e., by appropriately summing $\overline{A}_j$ to the final result),
    - therefore, the total computation for rearrangements and summation is, $\sum_{i=0}^{\ell} (\ell - i) \times |\Omega_i| \times |\mathcal{G}_{\ell-i,d-1}| \leq \ell \sum_{i=0}^{\ell} |\Omega_i| \times |\mathcal{G}_{\ell-i,d-1}| \underbrace{=}_{\text{P3}} \ell \times \mathcal{O}(\ell^{d-1}2^{\ell}) = \mathcal{O}(\ell^d 2^{\ell})$;
    - $\forall i, |\Omega_i|$ vectors need to be multiplied with $K_{\mathcal{G}_{\ell-i,d-1}}$, which is same computation as used in the $\overline{\mathbf{b}}_i$ step, therefore, it is $\mathcal{O}(\ell^{d+1}2^{\ell})$.

- For the $B_i$ step,
    - similar to $A_i$, $i$ rearrangements and summation (required for $\sum_{j\leq i} \mathcal{S}_{\mathcal{G}_{i,1},\Omega_j} \overline{B}_j \mathcal{S}_{\mathcal{G}_{\ell-j,d-1},\mathcal{G}_{\ell-i,d-1}}$), are performed simultaneously,
    - therefore, total computation for rearrangements and summation is, $\sum_{i=0}^{\ell} i \times |\mathcal{G}_{i,1}| \times |\mathcal{G}_{\ell-i,d-1}| \underbrace{\leq}_{\text{Eq. 6}} \ell \times \mathcal{O}(\ell^{d-1}2^{\ell}) = \mathcal{O}(\ell^d 2^{\ell})$.
    - the total MVM computation with $K_{\mathcal{G}_{\ell,1}}$ is same as for the $\overline{A}_i$ step, therefore its $\mathcal{O}(\ell^d 2^{\ell})$, as shown earlier.

- Finally, for the last re-arrangement in line 14, all updates are accumulated on $\mathbf{u}_i$. All $\mathbf{u}_i$ jointly are as large as $|\mathcal{G}_{\ell,d}|$, therefore $\mathcal{O}(\ell^{d-1}2^{\ell})$ computation is sufficient.

$\square$

**Batching-efficient Reformulation of Algorithm 1.**

In short, the main ideas behind iterative implementation can be summarized below:

- The recursions in Lines 8 and 12 can be batched together.
- Similarly, the recursion spawns many recursive multiplications with kernel matrices of the form $\mathcal{G}_{\ell',d'}$ for $0 \leq \ell' < \ell$ and $1 \leq d' < d$,

To achieve the above, we make following modifications:

- Re-organize computation of the Algorithm 1 and first loop over to compute $\overline{A}_i$ and $A_i$, followed by second loop over $\overline{B}_i$ and $B_i$.
- Notice since the computation of $\mathbf{u}_i$ depends on $B_i$, it implies that kernel-MVM with remaining dimensions need to be computed. Therefore, we run second loop over $\overline{B}_i$ and $B_i$ in the reverse order of dimensions compared Algorithm 1.
- At all computation steps, vectors are appropriately batched before multiplying with kernel matrices to improve efficiency.

### B.2 Simplicial Interpolation on Rectilinear Grids – Omitted Details

For detailed exposition of simplicial interpolation with rectilinear grids, we refer readers to Halton [11]. Briefly, the main idea is that each hypercube is partitioned into simplices, so the grid points themselves are still on the rectilinear grid (i.e., the corners of the hypercubes). For each grid point, the associated basis function takes value $1$ at the grid point, and is non-zero only for the simplices adjacent to that point, and takes value $0$ at the corner of those simplices. Therefore, it's linear on each simplex.

Concretely, for any $\mathbf{x} \in \mathbb{R}^d$, following steps are used to find basis function values and find grid points rectilinear grids that form the simplex containing $\mathbf{x}$.

1. Compute local coordinates $\mathbf{r}^{\mathbf{x}} = [\lambda(\mathbf{x}_1, \mathbf{s}_1), \cdots \lambda(\mathbf{x}_d, \mathbf{s}_d)]$, where $\forall i, \lambda(\mathbf{x}_i, \mathbf{s}_i) = \mathbf{x}_i - \lfloor \mathbf{x}_i / \mathbf{s}_i \rfloor \mathbf{s}_i$ and $\mathbf{s} \in \mathbb{R}^d$ is the spacing of rectilinear grid, i.e., distance between adjacent grid-points along all dimensions.

2. Sort local coordinates. We put $\mathbf{r}_{\mathbf{x}}$ in non-decreasing order, i.e., $\{\mathbf{o}_1, \mathbf{o}_2, \cdots, \mathbf{o}_d\} = \{1, 2, \cdots, d\}$ such that $1 \geq \mathbf{r}_{\mathbf{o}_1}^{\mathbf{x}} \geq \mathbf{r}_{\mathbf{o}_2}^{\mathbf{x}} \geq \cdots \geq \mathbf{r}_{\mathbf{o}_d}^{\mathbf{x}} \geq 0$ holds.

3. Compute interpolating basis values $\mathbf{v_x} \in \mathbb{R}^{d+1}$ as $\left[ 1 - \mathbf{r}_{\mathbf{o}_1}^{\mathbf{x}}, \mathbf{r}_{\mathbf{o}_1}^{\mathbf{x}} - \mathbf{r}_{\mathbf{o}_2}^{\mathbf{x}}, \mathbf{r}_{\mathbf{o}_2}^{\mathbf{x}} - \mathbf{r}_{\mathbf{o}_3}^{\mathbf{x}}, \cdots, \mathbf{r}_{\mathbf{o}_{d-1}}^{\mathbf{x}} - \mathbf{r}_{\mathbf{o}_d}^{\mathbf{x}}, \mathbf{r}_{\mathbf{o}_d}^{\mathbf{x}} \right]$.

4. Obtain neighbors by sorting the coordinates (columns) of the reference simplex (described below) so that they follow the same sorting order as the local coordinates. I.e., we sort the reference coordinates by the inverse sorting of the local coordinates.

Recall from the main text that there are several ways to partition the hypercube, i.e., several choices to build reference simplex. We build reference simplex $S \in \mathbb{R}^{d+1,d}$ by stacking $d+1$ row vectors, in particular, $\mathbf{1}^p \in \mathbb{R}^{1 \times d}$ vectors for $p \in [0, d]$ are stacked, where $\mathbf{1}^p \in \{0, 1\}^d$ has $d - p$ zeros followed by ones for the left-over entries.

## C  Experiments – Omitted Details and More Results

### C.1  Hyperparameters, optimization and data processing details

We run our experiments on Quadro RTX 8000 with $48$ GB of memory. For all experiments, we have used RBF kernel with separate length-scale for each dimension. For the optimization marginal log-likelihood, we use Adam optimizer with learning rate $0.1$ for 100 number of epochs. The optimization is stopped if no improvement is observed in the log-likelihood for 5 consecutive epochs.

The CG train and test tolerance are set to $1.0$ and $0.01$, which do not worsen perform in practice. Both CG pre-conditioning rank and maximum are 100. Our data is split in the ratio of $4:2:3$ to form train, validation and test splits. All UCI datasets are standardized using the training data to have zero mean and unit variance. For sparse-grid, we explore $\ell \in [2, 3, 4, 5]$ for Table 3. For dense-grid with simplicial interpolation, we explored grid points per dimension until we run out of memory.

### C.2  Another interpolation rule to apply sparse grids to large scale dataset

Recall that it's the relatively higher number of rectilinear grids used in a sparse grid that slows them down on large scale datasets. Analogous to the combination rule, we devise a new interpolation rule that only considers rectilinear grids in $\{\Omega_{\mathbf{l}} \mid \|\mathbf{l}\|_1 = \ell \mid (\ell \in \mathbf{l} \text{ OR } \ell - 1 \in \mathbf{l})\}$, i.e., $d^2/2 + d$ grids. Similar to combination interpolation technique, all grid interpolation weights are scaled by one by the total number of grids considered.

We focus on two large datasets with relatively higher dimensions, namely, Houseelectric and Airline datasets. Houseelectric has $\approx 2.05$ million data points with dimensionality $d = 11$. Similarly, Airline dataset has $\approx 5.92$ million data points with dimensionality $d = 8$. For Houseelectric, Sparse-grid performs comparable to Simplex-GP while being 3.95X faster. For the airline dataset, SKIP and SGPR go out-of-memory while Simplex-GP is slower by more than $4$ orders of magnitude. These results show that sparse grids when used with simplicial interpolation can be effective and efficient for large scale dataset.

**Table 3:** Test root-mean-square error (RMSE) and inference time on two large datasets with dimensions $d \geq 8$ and $n \geq 1M$. See text for more details on datasets. All numbers are averaged over 3 trials. OOM is out of memory. $\star$ number is taken from Kapoor et al. [13].

| Methods | Houseelectric | | Airline | |
|---|---|---|---|---|
| | RMSE | Time (in secs) | RMSE | Time (in secs) |
| SGPR | $0.067^\star$ | - | OOM | - |
| SKIP | OOM | - | OOM | - |
| Simplex-GP | **0.078** | 0.186 | 0.922 | 142.891 |
| Dense-grid | 0.170 | 0.263 | 0.892 | 0.413 |
| Sparse-grid | **0.088** | **0.047** | **0.832** | **0.003** |

## C.3 Sparse grid interpolation and GP inference for more synthetic functions.

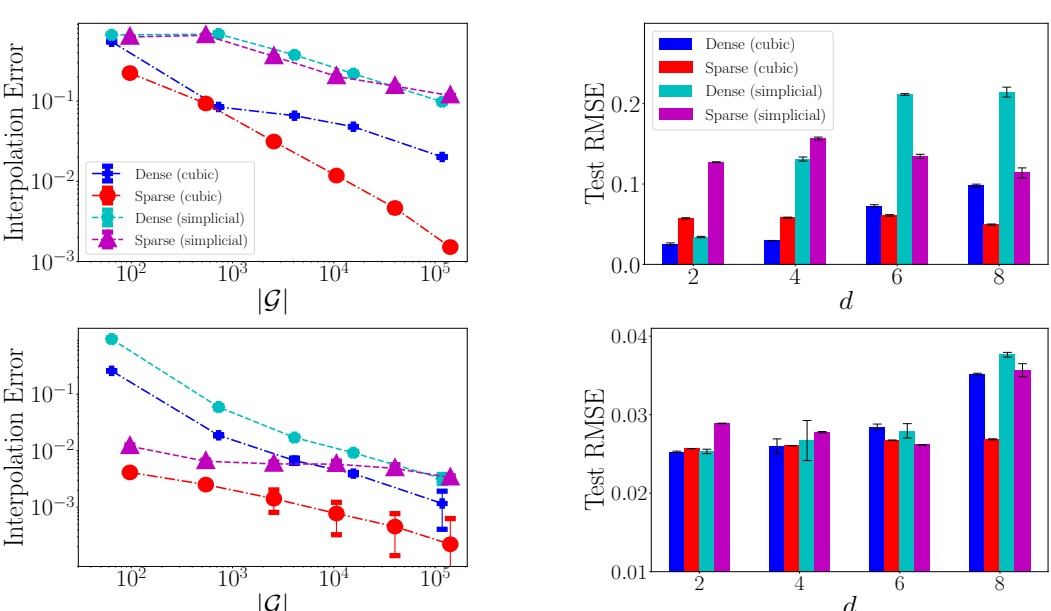

**Figure 6:** Additional results for two more synthetic functions, namely, *anisotropic and shifted* cosine (top panels) and *corner-peak* (bottom panels). Both left figures show that interpolation accuracy of sparse grids is comparable or superior for both interpolation schemes (i.e., cubic and simplicial). Furthermore, both right figures show that the advantage of sparse grids becomes more prominent as dimension increases. This effect is relatively less prevalent in bottom panel as the *corner-peak* function attain smaller values with increase in dimension. See text for precise function definitions.

Similar to section 4, we consider two more functions that are not isotropic (unlike $\cos(\|\mathbf{x}\|_1)$): a) *anisotropic and shifted* cosine function $f_d^{as}(\mathbf{x}) := \cos(2\pi w + \sum_{i=1}^d \mathbf{x}_i \mathbf{c}_i)$, b) *corner-peak* function $f_d^{cp}(\mathbf{x}) := (1 + \sum_{i=1}^d \mathbf{x}_i \mathbf{c}_i)^{-d-1}$. Both $w$ and $\mathbf{c}$ are selected randomly and remaining settings (i.e., noise, train and evaluation procedures) are same as in section 4.