# OpenReview forum: "Kernel Interpolation with Sparse Grids"
_NeurIPS.cc/2022/Conference — NeurIPS 2022 Accept_

### Official Review · Reviewer_GXVc · 2022-07-02

**Rating:** 8
**Confidence:** 4
**Soundness:** 4 excellent
**Presentation:** 4 excellent
**Contribution:** 3 good

**Summary:**

This paper uses sparse grids in structured kernel interpolation (SKI), which speeds up Gaussian process inference by replacing the kernal matrix with a kernel matrix evaluated at a structured set of points. In sparse grids the number of points grows faster with the dimension than in full grids typically used in SKI.

The two main contributions of the paper are (1) linear time algorithm for multiplying sparse grid kernel matrices with vectors that can be used in solving linear systems arising in GP inference via the conjugate gradient algorithm and (2) a simplicial interpolation scheme for approximating the full kernel matrix.


**Questions:**

Which class of kernels does the method apply to? Scattered remarks in the main paper and supplement seem to indicate that the kernel should be a product of one-dimensional stationary kernels but this is not made clear anywhere. In either Section 2 or 3 the reader should be explicitly told which kernels are considered in the paper.

**Limitations:**

One of the main limitations of the work is that although sparse grids are an improvement over full grids in this regard, the dimension that one can work in still remains fairly low (at most ten in this paper, I think). The authors are sufficiently upfront about this issue.

**Strengths And Weaknesses:**

Although the idea of replacing full grids with sparse grids is quite obvious and it is rather well known that, as opposed to orders of magnitudes more or something like that, one gains only a few extra dimensions with sparse grids (I appreciate that the paper does not attempt to hide this fact as is evident from line 80 and 272-273), I find this paper a very nice and useful contribution for the field. The paper is very well written and pleasant to read.

---

> ### Author Response · Authors · 2022-07-31
> **Response to reviewer GXVc**
>
> Thanks very much for your review and for highlighting our two key algorithmic contributions. We respond to your questions below.
>
> Question: “Which class of kernels does the method apply to?”
>
> Thanks – we will make this more explicit in the final version. Our fast algorithm applies to stationary product kernels, similar to most papers in the SKI literature. The product kernel assumption is critical since we recursively break down the kernel matrix across dimensions. The stationarity assumption is less critical – it allows us to apply fast FFT-based Toeplitz matrix-vector multiplication at the final step of recursion, yielding a runtime scaling near-linearly in the grid size. However, even without this, we would obtain an algorithm with a significantly faster runtime than the naive quadratic method.

---

### Official Review · Reviewer_vR8u · 2022-07-11

**Rating:** 5
**Confidence:** 3
**Soundness:** 3 good
**Presentation:** 3 good
**Contribution:** 3 good

**Summary:**

This study presents a way to improve the scalability of Structured kernel interpolation (SKI) for high dimensional dataset.  Specifically,  authors introduce the sparse grid inputs for constructing the inducing kernel Gram matrix $K_{G}$  in approximate kernel $W^{T}K_{G}W$. To use the sparse grid for SKI, they propose the MVM algorithm for the introduced sparse grid to have the near-linear computation for MVM, and the interpolation scheme for computing the $W$ (sparse weight). In this procedure, authors describe the properties of sparse grid and explain how the computational cost induced from sparse grid inputs could be much smaller compared to that of regular grid for SKI.
Through experiments, authors validate that using the sparse grid with the proposed schemes is computationally efficient while obtaining the comparable prediction performance.

**Questions:**

$\bullet $  Q1. For the scalable GP inference for high dimensional dataset, combining the random Fourier feature with fast food tricks enables the efficient inference in [1].  For the results in table 1, I am quite curious whether the proposed SKI framework could be comparable to other scalable GP inference scheme.

[1] A la Carte – Learning Fast Kernels, AISTATS 15


**Limitations:**

I think that if the validation procedure in experimental section is improved, this work could get a better evaluation.
Thus, I write some concerns about experiment section and some suggestions.

$\bullet $ For the experiment on synthetic dataset of $\cos({\vert{x}\vert}_{1})$, it seems that the more accurately the kernel is approximated in left figure ($d=6$), the more accurately it seems to predict in right figure ($d=6$). However, what if synthetic set are generated from different process, I am curious whether the effectiveness of sparse grid still remains (for generalization on a variety of datasets). If possible, what about applying the same experiment on some of the UCI datasets used in Table1?  If authors show the consistent results from the high dimensional UCI datasets, this method will be regarded as more powerful.

$\bullet $ For the UCI regression task, Table 1 does not report the standard deviation of the Test RMSE for 3 trials. Thus, I am not sure whether the proposed scheme results in stable training and prediction performance.  Also, for evaluating the GP regression task, it would be better to report the test negative log likelihood (nll), which is the widely used metric for GP regression task, as done [1].

$\bullet $ For regression task, comparing the training progress (xaxis: training time, yaxis: test rmse ) over varying number of sparse grid points $|\mathcal{G}|$  would be better to make sure that the sparse grid results in fast training for high dimensional dataset due to the reduced computational cost as shown in Figure 2 of [2].


[1] SKIing on Simplices: Kernel Interpolation on the Permutohedral Lattice for Scalable Gaussian Processes, ICML 21

[2] Product Kernel Interpolation for Scalable Gaussian Processes, AISTAST18




**Strengths And Weaknesses:**

$\textbf{Strengths}$

The existing SKI framework has been the scalable issue for high dimensional dataset, because the computational cost of MVM for the regular grid is $O(N4^{d} + |U| \log{|U|})$, and this implies that high dimensional data set (large $d$) increases the computational cost exponentially, and thus incurs the computational issue for employing the SKI. This work alleviates this scalable issue by inducing the sparse grid, and then proposing the corresponding MVM for sparse grid inputs. Since the introducing the sparse grids in the naïve manner does not result in the improvement of the computational cost as shown in Figure 1.(c), I think that making the near-linear computation for MVM computation with sparse grid is a quite novel part of this work.

$\textbf{Weaknesses}$

Although this work proposes a novel inference scheme for overcoming the issue of SKI, I think that the validation procedures of the proposed method in the experiment section is not sophisticated, and seems to a little bit weak to validate the generalization of the proposed work on a variety of dataset, as comparing the experiments validating the existing variants of SKI.
For further details, see Section Limitations below.

---

> ### Author Response · Authors · 2022-07-31
> **Response to reviwer vR8u**
>
> Thanks for the detailed review and for appreciating our algorithmic contribution of a fast kernel matrix-vector multiplication algorithm for sparse grids. Please find our responses to your questions below.
>
>
> Question: “For the scalable GP inference for high dimensional dataset, combining the random Fourier feature with fast food tricks enables the efficient inference in [1]. For the results in table 1, I am quite curious whether the proposed SKI framework could be comparable to other scalable GP inference schemes.”
>
> We have included SGPR as an (often state-of-the-art) baseline method for higher dimensional data sets, which is outside the SKI framework. This is consistent with many papers on kernel interpolation, including “SKIing on Simplices”, suggested in your response. Comparing with additional non-SKI methods, like random features methods, as you suggest, could be interesting.
>
> Question: On expanding the synthetic data experiment.
>
> Thanks for your suggestion here. Synthetic data allows us to vary only the dimension d while keeping the underlying function the same across dimensions, which is difficult to do for real-world datasets. We have done experiments with other functions and found results to be consistent as presented in the manuscript. We are happy to update the appendix with these results, to demonstrate the consistency of the finding.
>
>
> Question: On reporting RMSE standard deviations and negative log-likelihood in Table 1.
>
> Thanks for pointing this out – we are happy to add standard deviations to the results in the final version. The standard deviations are low – the maximum standard deviation is around 0.02 for all the results in Table 1, suggesting that our conclusions stand as reported. We also observe that both metrics (i.e., NLL and MSE) lead to a similar order of performance among different algorithms, for any given dataset. However, for completeness, we can report NLL results in the final version.  Both low standard deviation and consistent order of performance across different metrics can also be verified for the baseline algorithms via Table 2 of “SKIing on Simplices”.

---

> > ### Comment · Reviewer_vR8u · 2022-08-07
> > **Thanks authors for their response.**
> >
> > Thank authors for their response and answering for my questions. I confirm authors' claims that (1) the proposed method obtains the consistent results from the additional experiments using other synthetic functions, and (2) the proposed method has the small standard deviation that supports the superiority of the proposed method over baseline method. However, I feel missing that those results are not reported in the revised main or appendix manuscript that can be revised during the rebuttal procedure. I will keep my evaluation.

---

> > > ### Author Response · Authors · 2022-08-09
> > > **Additional results on synthetic functions and standard deviation for Table 1.**
> > >
> > > Thank you for responding to the rebuttal and reminding us that the submission can be updated.
> > >
> > > We have included both variance in Table 1 and more synthetic experiments in Appendix C.4. In summary, synthetic data experiments reinforce the original findings in the main section, i.e., sparse grid interpolation becomes more accurate with d > 4. Due to limited time, we could not do additional experiments for d=10 for the results presented in Figure 6.
> > >
> > >  We hope that these additional results address your points with more clarity. Thanks again.

---

### Official Review · Reviewer_9bwU · 2022-07-11

**Rating:** 6
**Confidence:** 3
**Soundness:** 3 good
**Presentation:** 3 good
**Contribution:** 3 good

**Summary:**

This manuscript proposes using sparse grids within the structured kernel interpolation framework to accelerate matrix vector products necessary when working with a kernelized Gaussian Process. Of particular note, sparse grids are preferable relative to dense grids as the dimension of the problem increases. A fast algorithm is provided for computing matrix vector products with the sparse grid based inducing point matrix and appropriate interpolation schemes are discussed. Finally, numerical results show the efficacy of the method for both synthetic and real world benchmark problems.

**Questions:**

- Is the plot in figure 1 (c) meant to show arithmetic operations or is this an actual timing experiment? "Complexity" is not a particularly descriptive y-axis. The statement "improves" suggests this is an actual experiment, but "complexity" suggests this is simply flop counting.

- Why are $\bar{\mathbf{a}}_i$ and $\bar{\mathbf{b}}_i$ lower case when they represent matrices? Also, maybe it would be clearer to make the recursive calls more closely match the algorithm input/output definition (technically, they can be fit into it by just calling the algorithm several times, but it does read a bit oddly given the input statement).

- Is there an explanation for the high variance in certain data points of Figure 2?

- Is Figure 2 showing the total MVM time or just the part with the kernel matrix (i.e., only Algorithm 1)? the text suggests this is an evaluation of Algorithm 1 and, therefore only $K$ is considered. However, by showing CG time the manuscript suggests it is the whole process including the interpolation matrices.

- In figure 3 what is the dense grid size? is it made to match the total number of points in the sparse grid or does each dimension match the maximum resolution. The results suggest the former (since the sparse grid interpolation is doing better), but this should be made clear.

- Is there a good explanation for why the sparse (simplicial) interpolation seems to be improving more slowly than simplicial interpolation using the dense grid?

**Limitations:**

- One limitation not discussed is the need to choose $\ell.$ While not dissimilar to parameters that appear in related methods (so I don't think this is a problem, per se), the manuscript does not provide much guidance on how select this parameter based on, e.g., the desired accuracy and that is a current limitation.

**Strengths And Weaknesses:**

Overall, I think that this paper makes a good contribution to the SKI framework. While sparse grids are certainly a well-studied and developed area, to the best of my knowledge they have not found use in this setting. Moreover, the development of a fast algorithm for applying the resulting inducing point matrix is key for this application. Accordingly, the idea + algorithm development represents a clear strength of the manuscript. Moreover, the exposition is relatively easy to read and the key points are well explained.

The main weakness of the paper is in its experimental evaluation of the method. Some of this may be relatively easy to address by clarifying certain details (see some specific questions below). Nevertheless, there are a few larger issues outlined below:

- The main focus of the paper seems to be on the efficiency of all parts of the SKI pipeline (i.e., both the inducing point matrix vector product and the interpolation). However, the most "comparative" experiment (table 1) simply looks at RMSE and comes off a bit as: "all these methods do reasonable in most cases and no one dominates the rest." While this is fine for showing the final result of the proposed method is comparable with other work, it does little to highlight the advantages (and/or potential disadvantages) of the proposed method relative to prior work. While there is a very small comparison in the appendix, given the focus of the paper it seems that balanced the numerical experiences more towards exploring the computational efficiency of the method would be preferable. For example, is the proposed method uniformly preferable? or when in say $d=2$ or $d=3$ does the practical efficacy of the FFT help dense grid methods win out? Where might one see the crossover to sparse grids being preferable? and more. Ideally the numerical experiments could help practitioners understand which method should be used in which settings (at least to a first approximation).

- The manuscript would benefit from a more clear breakdown between the interpolation time and the inducing point matrix multiplication. For example, which one is the dominant cost? This would help direct future work and also make it easier to evaluate the limitations where the manuscript suggests improvements to the interpolation are important (suggesting that this ends up being the dominant cost).

In summary, there is a clear algorithmic contribution in the manuscript that suggests the possibility of a practically efficient method in slightly larger dimensions than SKI is typically used (though there are still limits). Moreover, it seems likely that these idealized gains translate to practical improvements over prior methods. However, the experimental evaluation makes assessing the latter point somewhat challenging and the manuscript would benefit significantly by making a clearer case here.

Assorted notes:

- The definition of $V_i$ could be made a bit more clear; maybe a figure or more precise mathematical statement in the appendix?

- The fact that certain base case operations in Algorithm 1 are Topelitz and, therefore, can be accelerated using the FFT should probably be mentioned in the main text discussion of Algorithm 1. It is clear in the appendix, but important enough to warrant inclusion above.

- The authors note that SKI is typically infeasible for $d \geq 4$ and then immediately proceed to run a variant of SKI in higher dimensions. I think it is better to stick to the more general "scaling" arguments rather than trying to draw somewhat arbitrary boundaries between methods based on the dimension.

- Proposition 2 is styled different than the other theoretical statements (e.g., Proposition 1 and Theorem 1). While either is fine, the styling should be consistent.

- The related work section focuses mainly on SKI type methods. While that being the focus is certainly appropriate given the contribution of this manuscript, given that the problem being solved is GPs and not "improve SKI" the manuscript may want to allude to the fact that there are other "styles" of algorithms that are also being developed to try and scale to larger dimensions (though they all certainly have their own limitations and pros/cons relative to SKI). For example, one common strategy is the use of rank-structured factorizations (as with SKI these are most effective for $d\leq 3$ but there has been work on "scaling up" to more moderate dimensions:

[Yang, Changjiang, Ramani Duraiswami, Nail A. Gumerov, and Larry Davis. "Improved fast gauss transform and efficient kernel density estimation." In Computer Vision, IEEE International Conference on, vol. 2, pp. 464-464. IEEE Computer Society, 2003.]

[Ambikasaran, Sivaram, Daniel Foreman-Mackey, Leslie Greengard, David W. Hogg, and Michael O’Neil. "Fast direct methods for Gaussian processes." IEEE transactions on pattern analysis and machine intelligence 38, no. 2 (2015): 252-265.]

[Ryan, John P., Sebastian E. Ament, Carla P. Gomes, and Anil Damle. "The fast kernel transform." In International Conference on Artificial Intelligence and Statistics, pp. 11669-11690. PMLR, 2022.]

- It seems ref 24 is a duplicate of 23? I believe this is just one paper from 2005 unless there is another reference I am unfamiliar with that is meant by the second appearance.

---

> ### Author Response · Authors · 2022-07-31
> **Response to reviwer 9bwU**
>
> Many thanks for the detailed comments and questions. We appreciate the suggestions on presentation in your “assorted notes” section and will address them in the final version.
>
> We agree that further experiments on scalability, especially with respect to dimension, would strengthen the work. We also agree that including a more clear breakdown of the MVM complexities of W and K matrices, in terms of both time and space complexity, would be valuable and illustrate which matrix is the bottleneck in which parameter regimes. We anticipate that for d=2 or d=3, dense grids will remain preferable over sparse grids, due to the practical efficiency of FFT-based kernel matrix-vector multiplication routines for these grids. Sparse grids however become preferable in moderate dimensions. E.g., in Figure 3, we begin seeing the advantages of the sparse grid around d = 6.
>
> Question: “Is the plot in figure 1 (c) meant to show arithmetic operations or is this an actual timing experiment?.”
>
> We will clarify this. 1(c) is simply counting flops. It is not an empirical result.
>
>
> Question: “Why are a_i and b_i  lower case when they represent matrices? Also, maybe it would be clearer to make the recursive calls more closely match the algorithm input/output definition (technically, they can be fit into it by just calling the algorithm several times, but it does read a bit oddly given the input statement).”
>
> Thanks. We will fix this notation, and clarify the recursive calls.
>
> Question: “Is there an explanation for the high variance in certain data points of Figure 2?”
>
> Some measurements are tiny (note the log scale on the y-axis), so the error bars are typically small in absolute terms. We also only ran 8 trials – we will increase this.
>
> Question: “Is Figure 2 showing the total MVM time or just the part with the kernel matrix (i.e., only Algorithm 1)?”
>
> It just focuses on the part with the kernel matrix (i.e., Algorithm 1). However, as detailed in the caption, we attempt to predict the typical runtime required for all kernel matrix-vector products in a typical run of CG using the formula CG time = pre-processing time + 50 times MVMs. We then empirically measure these runtimes. Pre-processing time e.g., includes constructing the full kernel matrix for the baseline algorithm, or constructing the 1-dimensional kernel matrices at different resolutions in our fast algorithm.
>
> Question:  “In figure 3 what is the dense grid size? is it made to match the total number of points in the sparse grid or does each dimension match the maximum resolution.”
>
> Yes, it is the former. We tried to match the sizes of both dense and sparse grids while ensuring that the dense grid always had at least as many points as the sparse grid, to give a fair comparison. Precisely, (d, dense grid size, sparse grid size) tuples are  (2, 144, 129) , (4, 1296, 796), (6, 4096, 2561),  (8, 6561, 6401),  (10, 59049, 13441).
>
>
> Question:  “Is there a good explanation for why the sparse (simplicial) interpolation seems to be improving more slowly than simplicial interpolation using the dense grid?”
>
> We are not sure if this is a general phenomenon or an artifact of some part of our experiment setup (e.g., the slight mismatch in grid sizes), and will need to investigate further.

---

### Official Review · Reviewer_pbvp · 2022-07-12

**Rating:** 7
**Confidence:** 4
**Soundness:** 3 good
**Presentation:** 3 good
**Contribution:** 3 good

**Summary:**

The manuscript proposes a methodology to speed up structured kernel interpolation Gaussian processes by using a grid with  a number of grid points that does not scale exponentially in the number of data dimension. The sparse grid approach is borrowed from computational mathematics and used to render Gaussian processes more scalable. A number of experiments on datasets with d>>4 illustrates the potential of the method.

**Questions:**

* The grid point density is not evenly distributed over the input space. Does this imply a preprocessing to the data to concentrate the relevant parts close to the coordinate axis where the density is higher?
* In Figure 1, you could align the color of the complexity panel and the grid visualisation panels.
* The term "nearly linear" should be properly defined or changed.

**Limitations:**

Yes.

**Strengths And Weaknesses:**

1) Clarity
The paper is well written. The notation language and figures are concise and well accessible.
Two typos: Line 85: scalibility, Line 287: Sentence needs to be revised.

2) Originality
The proposed (thinner) grid seems to not have been applied to scalable Gaussian process inference.

3) Significance
The method is a proper extension of SKI and is not specific to a particular covariance or likelihood function and can be used in a wide range of inference schemes. As demonstrated in the experiments, SKI is applicable to moderate dimensions up to 10 which was previously challenging.

4) Reproducibility
As the code for the experiments is available on https://anonymous.4open.science/r/skisg-485E/README.md, the results should be simple to reproduce. I'm assuming that the code will be shared at a more accessible location upon acceptance.

5) Empirical analysis
The experimental section reports a number of results on runtime and memory footprint and demonstrates the utility of the method.

---

> ### Author Response · Authors · 2022-07-31
> **Response to reviewer pbvp**
>
> Thanks for your appreciation of several aspects of the paper and for your helpful suggestions. Responses to your questions are below:
>
>
> Question: “The grid point density is not evenly distributed over the input space. Does this imply a preprocessing  the data to concentrate the relevant parts close to the coordinate axis where the density is higher?”
>
> Yes – it is correct that the grid density is not evenly distributed. However, the data is not processed to concentrate close to the coordinate axis and/or higher grid density regions. We suspect that for moderate dimensional datasets, finding an alignment of the data that significantly increased average proximity to the grid points would be difficult. However, it would be interesting to investigate this.
>
>
> Question: “The term "nearly linear" should be properly defined or changed.”
>
> Thanks, we will clarify "nearly linear". A sparse grid in d-dimensions with resolution l contains G_{l,d} = l^{d-1} x 2^l points (see Prop. 1), and our kernel matrix vector multiplication algorithm has runtime O(l^{d+1}*2^l) = O(G_{l,d}*log(G_{l,d})^{d+1}) . Thus, up to polylogarithmic factors, the runtime scales linearly in G_{l,d}, which is why we use the term ‘nearly linear’.

---

### Author Response · Authors · 2022-07-30
**Common note for all reviwers.**

We thank all the reviewers for their in-depth reviews and valuable feedback. We are happy that reviewers recognize the significance of our contributions, and felt they were well supported by evidence. We appreciate the suggestions on the presentation and points of clarification and will address them in the final version. We respond to specific questions/comments by reviewers next to their feedback.

---

### Meta-Review · Area_Chair_PRYh · 2022-08-26

**Recommendation:** Accept
**Confidence:** Certain

**Metareview:**

All reviewers found this paper relevant for the conference, original, and well written. All four reviewers recommended accepting the paper. For the camera-ready, please go through the reviewer comments in detail and check that you have addressed the remaining concerns regarding clarity and improving presentation.

For the camera-ready version, you also need to fix the font issue in your paper. In the current version, the font/text size is not what it should be (most likely due to some package clash).

**Award:**

No

---

### Decision · Program_Chairs · 2022-09-14

Accept